# Glia actively sculpt sensory neurons by controlled phagocytosis to tune animal behavior

Stephan Raiders[1,2], Erik Calvin Black[1], Andrea Bae[3,4†], Stephen MacFarlane[5], Mason Klein[5], Shai Shaham[3], Aakanksha Singhvi[1,2,6,7]*

[1]Division of Basic Sciences, Fred Hutchinson Cancer Research Center, Seattle, United States; [2]Molecular and Cellular Biology Graduate Program, University of Washington, Seattle, United States; [3]Laboratory of Developmental Genetics, The Rockefeller University, New York, United States; [4]Cellular Imaging Shared Resources, Fred Hutchinson Cancer Research Center, Seattle, United States; [5]Department of Physics and Department of Biology, University of Miami, Coral Gables, United States; [6]Department of Biological Structure, University of Washington School of Medicine, Seattle, United States; [7]Brotman Baty Institute for Precision Medicine, Seattle, United States

*For correspondence:
asinghvi@fredhutch.org

Present address: †Dominick P. Purpura Department of Neuroscience, Albert Einstein College of Medicine, Bronx, United States

Competing interests: The authors declare that no competing interests exist.

**Abstract** Glia in the central nervous system engulf neuron fragments to remodel synapses and recycle photoreceptor outer segments. Whether glia passively clear shed neuronal debris or actively prune neuron fragments is unknown. How pruning of single-neuron endings impacts animal behavior is also unclear. Here, we report our discovery of glia-directed neuron pruning in *Caenorhabditis elegans*. Adult *C. elegans* AMsh glia engulf sensory endings of the AFD thermosensory neuron by repurposing components of the conserved apoptotic corpse phagocytosis machinery. The phosphatidylserine (PS) flippase TAT-1/ATP8A functions with glial PS-receptor PSR-1/PSR and PAT-2/α-integrin to initiate engulfment. This activates glial CED-10/Rac1 GTPase through the ternary GEF complex of CED-2/CrkII, CED-5/DOCK180, CED-12/ELMO. Execution of phagocytosis uses the actin-remodeler WSP-1/nWASp. This process dynamically tracks AFD activity and is regulated by temperature, the AFD sensory input. Importantly, glial CED-10 levels regulate engulfment rates downstream of neuron activity, and engulfment-defective mutants exhibit altered AFD-ending shape and thermosensory behavior. Our findings reveal a molecular pathway underlying glia-dependent engulfment in a peripheral sense-organ and demonstrate that glia actively engulf neuron fragments, with profound consequences on neuron shape and animal sensory behavior.

## Introduction

To interpret its environment accurately and respond with appropriate behaviors, an animal's nervous system needs to faithfully transmit information from the periphery and through neuron–neuron contacts within the neural network. Precision in this information transfer and processing depends partly on neuron-receptive endings (NREs), specialized subcellular structures where a neuron receives input from either the external environment or other neurons (*Bourne and Harris, 2008*; *Harms and Dunaevsky, 2007*; *Shaham, 2010*; *Singhvi et al., 2016*). In the peripheral nervous system (PNS), sensory NREs house the sensory transduction machinery and appropriate NRE shape is important for sensory information capture. In the central nervous system (CNS), the size and number of interneuron NREs (dendritic spines) help determine the connectome and thereby the path of information

**eLife digest** Neurons are tree-shaped cells that receive information through endings connected to neighbouring cells or the environment. Controlling the size, number and location of these endings is necessary to ensure that circuits of neurons get precisely the right amount of input from their surroundings.

Glial cells form a large portion of the nervous system, and they are tasked with supporting, cleaning and protecting neurons. In humans, part of their duties is to 'eat' (or prune) unnecessary neuron endings. In fact, this role is so important that defects in glial pruning are associated with conditions such as Alzheimer's disease. Yet it is still unknown how pruning takes place, and in particular whether it is the neuron or the glial cell that initiates the process.

To investigate this question, Raiders et al. enlisted the common laboratory animal *Caenorhabditis elegans*, a tiny worm with a simple nervous system where each neuron has been meticulously mapped out. First, the experiments showed that glial cells in *C. elegans* actually prune the endings of sensory neurons. Focusing on a single glia-neuron pair then revealed that the glial cell could trim the endings of a living neuron by redeploying the same molecular machinery it uses to clear dead cell debris. Compared to this debris-clearing activity, however, the glial cell takes a more nuanced approach to pruning: specifically, it can adjust the amount of trimming based on the activity load of the neuron.

When Raiders et al. disrupted the glial pruning for a single temperature-sensing neuron, the worm lost its normal temperature preferences; this demonstrated how the pruning activity of a single glial cell can be linked to behavior.

Taken together the experiments showcase how *C. elegans* can be used to study glial pruning. Further work using this model could help to understand how disease emerges when glial cells cannot perform their role, and to spot the genetic factors that put certain individuals at increased risk for neurological and sensory disorders.

---

transfer (*Bargmann and Marder, 2013*; *Eroglu and Barres, 2010*; *Nimchinsky et al., 2002*). While remodeling of NRE shape has been suggested to be important for experiential learning and memory (*Bourne and Harris, 2008*; *Harms and Dunaevsky, 2007*), directly correlating these subcellular changes with animal behavior has been challenging.

Glia are a major cell type of the nervous system and approximate neurons in number (*von Bartheld et al., 2016*). They have been proposed to actively modulate development, homeostasis, and remodeling of neural circuits, and are thought to influence NRE shape and numbers (*Allen and Eroglu, 2017*; *Stogsdill and Eroglu, 2017*; *Zuchero and Barres, 2015*). One mechanism by which glia may do so is by engulfment of neuron fragments, including NREs (*Freeman, 2015*; *Schafer and Stevens, 2013*; *Wilton et al., 2019*). Aberrant neuron fragment uptake by glia is implicated in neurodevelopmental as well as neurodegenerative diseases, including Alzheimer's dementia, autism, and epilepsy (*Chung et al., 2015*; *Henstridge et al., 2019*; *Neniskyte and Gross, 2017*; *Schafer and Stevens, 2013*; *Vilalta and Brown, 2018*; *Wilton et al., 2019*).

Fundamental questions about the roles and mechanisms of glia-dependent phagocytosis remain open. Whether glia initiate engulfment or passively respond to neuron shedding is unclear. Furthermore, correlating glia-dependent remodeling at single synapse or NREs with changes in animal behavior remains challenging in most systems (*Koeppen et al., 2018*; *Wang et al., 2020*). Also, glial engulfment mechanisms have been primarily dissected in the context of injury or development, and their impact on adult neural functions remains less understood. Finally, whether glia-dependent engulfment occurs in the peripheral nervous system (PNS) or dictates normal sensory functions has not been extensively explored.

The nervous system of the adult *Caenorhabditis elegans* hermaphrodite comprises 302 neurons and 56 glial cells (*Singhvi and Shaham, 2019*; *Sulston et al., 1983*; *White et al., 1986*). These arise from invariant developmental lineages, form invariant glia–neuron contacts, and each neuron performs defined functions to enable specific animal behaviors. These features allow single-cell and molecular analyses of individual glia–neuron interactions with exquisite precision (*Singhvi et al., 2016*; *Singhvi and Shaham, 2019*).

Here, we describe our discovery that the *C. elegans* AMsh glial cell engulfs NRE fragments of the major thermosensory neuron of the animal, AFD. Thus, this critical glial function is conserved in the nematode and across sense-organ glia. We find that engulfment requires the phospholipid transporter TAT-1/ATP8A, α-integrin PAT-2, and glial phosphatidylserine receptor PSR-1. PSR-1 engages a conserved ternary GEF complex (CED-2/CrkII, CED-5/DOCK180, CED-12/ELMO1) to activate CED-10/Rac1 GTPase. The actin remodeling factor WSP-1/nWASp, a known effector of CED-10, acts in AMsh glia to regulate engulfment. We also show that glial engulfment rates are regulated by temperature and track AFD neuron activity. Importantly, glial CED-10/Rac1 acts downstream of neuron activity, and CED-10 expression levels dictate NRE engulfment rates. Finally, perturbation of glial engulfment leads to defects in AFD–NRE shape and associated animal thermosensory behavior. Our studies show that glia actively regulate engulfment by repurposing components of the apoptotic phagocytosis machinery. Importantly, while cell corpse engulfment is an all-or-none process, glia-dependent engulfment of AFD endings can be dynamically regulated. We propose that other glia may similarly deploy regulated phagocytosis to tune sensory NREs and synapses, and to dynamically modulate adult animal behaviors.

## Results

### *C. elegans* glia engulf fragments of the AFD–NRE

Glia of the nematode *C. elegans* share molecular, morphological, and functional features with vertebrate sense-organ glia and astrocytes (*Bacaj et al., 2008a*; *Katz et al., 2018*; *Katz et al., 2019*; *Lee et al., 2021*; *Singhvi and Shaham, 2019*; *Wallace et al., 2016*). In previous studies, we established the AMsh glia–AFD neuron pair as a tractable experimental platform to define molecular mechanisms of single glia–neuron interactions (*Singhvi et al., 2016*; *Singhvi and Shaham, 2019*; *Wallace et al., 2016*). The AFD–NRE comprises ~45 actin-based microvilli and a single microtubule-based cilium that are embedded in the AMsh glial cell. An adherens junction between the AFD–NRE base and the AMsh glial cell isolates this glia–NRE compartment (*Figure 1A, B*; *Doroquez et al., 2014*; *Perkins et al., 1986*).

Upon imaging fluorescently labeled AFD–NREs in transgenic animal strains, we consistently observed labeled fragments disconnected from the neuron (*Figure 1C, C′*, *Video 1*). Our previous reconstructions-based FIB-SEM serial section data had also revealed AFD–NRE fragments disconnected from the rest of the AFD neuron (marked yellow, *Video 1*) in *Singhvi et al., 2016*. We examined this further using two-color imaging, which revealed that many of these fragments reside within the AMsh glial process and cell body (*Figure 1D–F′*, *Video 2*). To confirm that these glial puncta do not reflect spurious reporter protein misexpression in glia but rather derive from the AFD, we ablated AFD neurons early in larval development and looked for puncta on the first day of adulthood. Upon ablation of one of the two bilateral AFD neurons by laser microsurgery in first larval stage (L1) animals, fragment formation was blocked on the operated side, but not on the unoperated side, or in mock-ablated animals (*Figure 1G, H*). Similar results were seen with stochastic genetic ablation of AFD using the pro-apoptotic BH3-domain protein EGL-1, expressed using an embryonic AFD-specific promoter (*Figure 1I*). We conclude, therefore, that AMsh glia engulf fragments of the AFD–NRE in *C. elegans*.

3D super-resolution microscopy studies revealed that the average size of AFD-derived glial puncta is 541 ± 145 nm along their long (yz) axis (*Figure 2A*). These fragments are an order of magnitude smaller than recently described exophers extruded from neurons exposed to cellular stress (~3.8 μm in diameter) and larger than ciliary extracellular vesicles (~150 nm) (*Chung et al., 2013*; *Melentijevic et al., 2017*; *Wang et al., 2014*). This size is of the same order of magnitude as the sizes of individual AFD–NRE microvilli or cilia as measured by electron microscopy (*Figure 2B*, *Figure 2—figure supplement 1A*) and (*Doroquez et al., 2014*).

### AMsh glia engulfment of AFD–NREs occurs in adults

Engulfment of neuronal fragments by glia has been suggested to refine neuronal circuit connectivity during neural development (*Chung et al., 2013*; *Wilton et al., 2019*). Post development, glial engulfment is thought to regulate animal behaviors and memory (*Koeppen et al., 2018*; *Wang et al., 2020*). To determine when *C. elegans* AMsh glia initiate engulfment of AFD–NRE

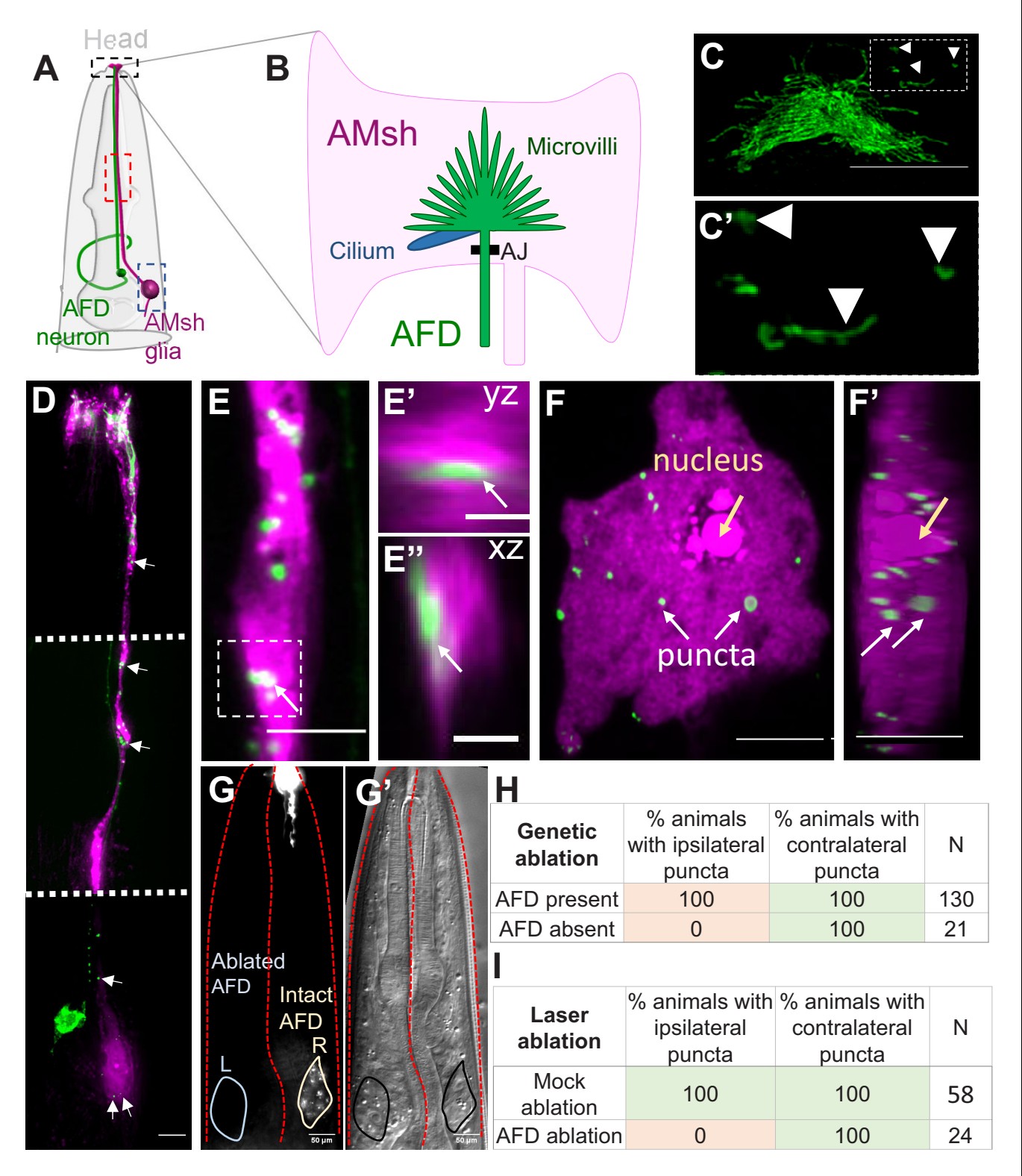

**Figure 1.** AMsh glia contain AFD–NRE-labeled puncta. (A) Schematic of the *C. elegans* head region depicting AFD neuron and AMsh glial cell body and processes. Anterior is to the top. Black box: zoomed in (B, C); red box region zoomed in (E); blue box zoomed in (F). (B) The AMsh glia's anterior ending ensheathes AFD–NRE dendrite, which comprises ~45 microvilli (green) and a single cilium (blue). AJ: adherens junction between AMsh glia and AFD neuron. (C, C') $P_{SRTX-1b}$:SRTX-1:GFP specifically labels AFD–NRE microvilli. Arrows indicate microvilli fragments disconnected from the main AFD–NRE structure, zoomed in (C'). Anterior is to the top. Scale bar: 5 μm. (D–F') Fluorescence micrograph of AMsh glia (magenta) show AFD–NRE puncta

*Figure 1 continued on next page*

*Figure 1 continued*

throughout the cell (D) including the process (E) and soma (F). Image in (D) is a composite of three exposure settings of a single animal, stitched where indicated by dotted white line. Orthogonal slices of AMsh glial process (E′, E′′, scale bar: 2 µm) and cell body (F′) show AFD–NRE fragments completely within AMsh glia. Scale bar: 5 µm. (G, G′) Day 1 adult animal with left AFD neuron ablated by laser microsurgery during L1 larval stage. Left AMsh soma (blue outline) lacks AFD–NRE fragments, right AMsh soma (green outline) contains fragments. (G) Fluorescence micrograph, (G′) differential intereference contrast (DIC) microscopy image. (H, I) Quantification of puncta in ipsilateral and contralateral AMsh glial cell soma with AFD neurons ablated by laser (H) or genetically (I). N: number of animals assayed; NRE: neuron-receptive ending.

The online version of this article includes the following source data for figure 1:

**Source data 1.** Raw data for *Figure 1H, I*.

---

fragments, we counted engulfed NRE puncta at different life stages. We found that these puncta are rarely found in embryos or early larval stages, but are easily detected in L4 larvae and increase in numbers during adulthood (*Figure 2C, D*). Thus, consistent with L1 laser ablation studies (*Figure 1G–I*), engulfment of AFD–NREs by glia occurs after development of the AFD–NRE is largely complete.

We found that ~65% of 1-day old adult animals expressing the AFD–NRE-specific *gcy-8*:GFP raised at 20˚C have AMsh glia containing >10 puncta, and another ~32% of animals have 1–9 puncta/glia (n = 171) (*Figure 2C*) (see Materials and methods for binning details). The AMsh glial cell of 1-day-old adults has on average 14 ± 1 puncta (n = 78) (*Figure 2D*). Using time-lapse microscopy, we found that individual puncta separate from the NRE at a frequency of 0.8 ± 0.3 events/min and travel at 1.05 ± 0.1 µm/s down the glial process towards the cell body, consistent with motor–protein-dependent retrograde trafficking (quantifications of videos from n = 5 animals) (*Figure 2—figure supplement 1B*, *Videos 1* and *2*; *Maday et al., 2014*; *Paschal et al., 1987*). Finally, age-matched animals raised at different cultivation temperatures differ in glia puncta accumulation (*Figure 2E*).

## AMsh glia engulf AFD–NRE microvilli but not cilia

AFD–NREs comprise multiple microvilli and a single cilium (*Figure 1B*). The size of puncta we observed (541 ± 145 nm, *Figure 2A*) was similar to the diameters of both the microvilli (214 ± 30 nm) (*Figure 2B*, *Figure 2—figure supplement 1A*) and AFD cilium (264 ± 13 nm) (*Doroquez et al., 2014*), precluding easy inference of the source of these puncta. To distinguish which organelle was engulfed, we undertook two approaches. First, we labeled each organelle with specific fluorescent tags and examined uptake by AMsh glia. To probe microvilli, we examined transgenic animals labeled with either of four AFD-micro-villi-specific proteins with fluorescent tags, SRTX-1, GCY-8, GCY-18, and GCY-23 (*Colosimo et al., 2004*; *Inada et al., 2006*). We found that all four transgenic strains consistently show fluorescent puncta in glia (*Figure 1*, *Figure 3A*). Time-lapse microscopy of one of these (P$_{srtx-1}$:SRTX-1:GFP) also revealed that fragments originate from the AFD–NRE microvilli (*Figure 2—figure supplement 1B*, *Videos 1* and *2*). To label cilia, we generated transgenic animals with the ciliary protein DYF-11/TRAF31B1 fluorescently tagged and expressed under an AFD-specific promoter and confirmed that P$_{AFD}$:DYF-11:GFP localizes to AFD cilia (*Figure 3B*). However, we found no DYF-11:GFP puncta in AMsh glia (*Figure 3A*).

In a complementary approach, we examined mutants lacking either microvilli or cilia. The

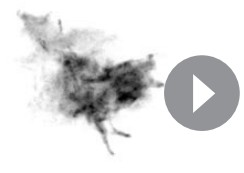

0 sec

**Video 1.** Dissociation of AFD–NRE fragments. Movie of an animal's AFD–NRE, labeled with GFP and imaged in vivo at 7 frames/s, shows fragments blebbing at regular intervals. NRE: neuron-receptive ending.
https://elifesciences.org/articles/63532#video1

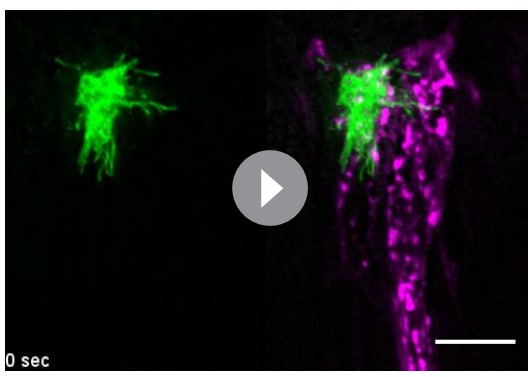

**Video 2.** AFD–NRE fragments are engulfed by AMsh glia. Movie of an animal's AFD–NRE (green) and AMsh glia (magenta) imaged in vivo at 7 frames/s shows fragments blebbing at regular intervals. NRE: neuron-receptive ending.
https://elifesciences.org/articles/63532#video2

development of AFD, including its microvilli (but not cilia), requires the terminal selector transcription factor TTX-1/Otx1/orthodenticle (*Hobert, 2016*; *Satterlee et al., 2001*). We found that *ttx-1(p767)* mutants lack AFD–NRE puncta in AMsh glia (*Figure 3C–E*). Cilia development requires the IFT-B early assembly proteins DYF-11/TRAF31B1 and OSM-6/IFT52. Both are expressed in most, if not all, ciliated neurons, and mutations in the respective genes exhibit defective amphid cilia (*Bacaj et al., 2008a*; *Collet et al., 1998*; *Kunitomo and Iino, 2008*; *Li et al., 2008*; *Perkins et al., 1986*; *Starich et al., 1995*). In contrast to *ttx-1* mutants, glia puncta were present in animals mutant for either *dyf-11(mn392)* or *osm-6(p811)* (*Figure 3C–E*). In fact and on the contrary, we found that *dyf-11* cilia-defective mutants accumulate more glial puncta that wild-type animals (*dyf-11:* 38 ± 3 puncta, n = 27 vs. *wild type:* 14 ± 1, n = 78, *Figure 3C, E*; and a larger fraction of *dyf-11* and *osm-6* mutants exhibit >10 puncta/glia [*dyf-11:* 95%, n = 61 animals, *osm-6:* 100% animals, n = 82, vs. wild type: 65%, n = 171]; *Figure 3D*). This indicates that cilia are likely not the primary source of glia puncta.

Data from all these approaches taken together suggest that that the observed puncta in AMsh glia derive from AFD–NRE microvilli as the primary, if not sole, source.

## The phospholipid transporter TAT-1 regulates glial engulfment

What molecular mechanism drives AFD–NRE microvilli engulfment? In other contexts, neurons expose the membrane phospholipid phosphatidylserine (PS) on the outer leaflet of the plasma membrane as a signal for glial phagocytosis (*Hakim-Mishnaevski et al., 2019*; *Li et al., 2020*; *Nomura-Komoike et al., 2020*; *Raiders et al., 2021*; *Scott-Hewitt et al., 2020*). However, the underlying molecular mechanisms that regulate this exposure in neurons are unclear. Apoptotic corpse phagocytosis, including in *C. elegans*, is also mediated by PS exposure (*Figure 4A*). PS exposure in apoptotic cells is promoted partially by the Xkr8 factor CED-8, which is cleaved by the caspase CED-3 to promote PS presentation for cell corpse phagocytosis (*Bevers and Williamson, 2016*; *Wang et al., 2007*). However, mutations in neither *ced-8* (*Figure 4B*) nor *ced-3* (data not shown) affect glial NRE uptake. Likewise, mutations in *scrm-1*, encoding a scramblase-promoting PS exposure (*Wang et al., 2007*), only mildly decrease AFD–NRE engulfment (*Figure 4B*). However, a presumptive null mutation in *tat-1*, an ortholog of mammalian translocase ATP8A required for PS sequestration to the plasma membrane inner leaflet (*Andersen et al., 2016*), results in increased apoptotic cell corpse engulfment (*Darland-Ransom et al., 2008*; *Hong et al., 2004*) and AFD–NRE engulfment (*Figure 4B, E*). Thus, common and context-specific mechanisms control apoptotic and NRE engulfment. Importantly, re-expression of wild-type *tat-1* cDNA under an AFD-specific promoter fully rescues the *tat-1* engulfment defect (*Figure 4B*). We conclude that cell-autonomous function of the PS-flippase TAT-1 in the AFD neuron regulates engulfment of AFD–NRE fragments by AMsh glia.

## The PS-receptor PSR-1 acts with the transthyretin TTR-52 to mediate glial engulfment

How is PS on the AFD membrane recognized by AMsh glia? To address this question, we examined mutants in receptors required for *C. elegans* apoptotic cell engulfment (*Figure 4A*). CED-1/Draper/MEGF10 is required for removal of neuron debris in many contexts (*Cherra and Jin, 2016*; *Mangahas and Zhou, 2005*; *Nichols et al., 2016*), including by glia in other species (*Chung et al., 2013*; *Freeman, 2015*; *Hamon et al., 2006*; *Raiders et al., 2021*). Surprisingly, two independent *ced-1* loss-of-function alleles do not block NRE fragment uptake (*Figure 4C*). Similarly, disrupting

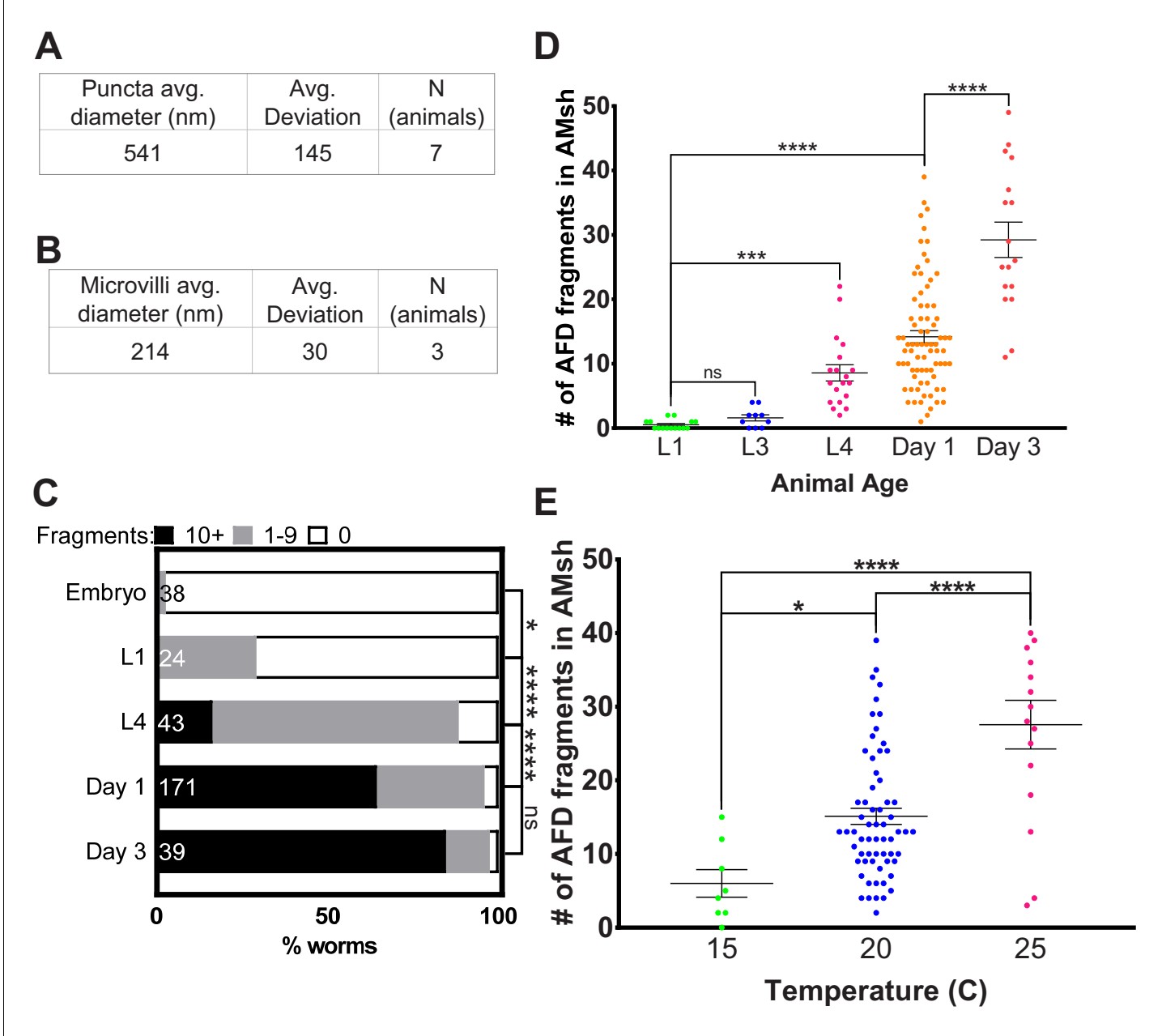

**Figure 2.** AMsh glia puncta engulf AFD–NRE. (**A**) Quantification of average puncta diameter within AMsh glial cell soma. (**B**) Quantification of average AFD–NRE microvilli diameter from electron micrographs. (**C**) Population scores of wild-type animals with AFD–NRE-labeled fragments within AMsh soma at different developmental stages. X-axis: percent animals with fragments. Y-axis: developmental stage. Puncta numbers are quantified into three bins (≥10 fragments, black bar), (1–9 fragments, gray bar), (0 fragments, white bar). N: number of animals. Statistics: Fisher's exact test. *p<0.05, **p<0.005, ***p<0.0005, ****p<0.00005, ns = p>0.05. See Materials and methods for details. (**D**) Quantification of AFD–NRE-labeled fragments within AMsh soma at different developmental stages. X-axis: developmental stage. Y-axis: number of puncta per AMsh glial cell soma. Median puncta counts and N (number of animals): L1 larva (0.5 ± 0.2 puncta, n = 15 animals), L3 larva (1.6 ± 0.5 puncta, n = 10 animals), L4 larva (8.6 ± 1.2 puncta, n = 19 animals), day 1 adult (14.1 ± 1 puncta, n = 78 animals), and day 3 adult (29.2 ± 3 puncta, n = 17 animals). Statistics: one-way ANOVA w/ multiple comparison. *p<0.05, **p<0.005, ***p<0.0005, ****p<0.00005, ns = p>0.05. (**E**) Average number of fragments in animals cultivated at 15℃, 20℃, or 25℃. Refer (**D**) for data presentation details. Median puncta counts and N (number of animals): 15℃ (6 ± 2 puncta, n = 8 animals), 20℃ (14.1 ± 1 puncta, n = 78 animals), and 2 5℃ (27.6 ± 3 puncta, n = 16 animals). NRE: neuron-receptive ending.

The online version of this article includes the following source data and figure supplement(s) for figure 2:

**Source data 1.** AMsh glia puncta engulf AFD–NRE.
**Figure supplement 1.** AMsh glia engulf AFD–NRE fragments.

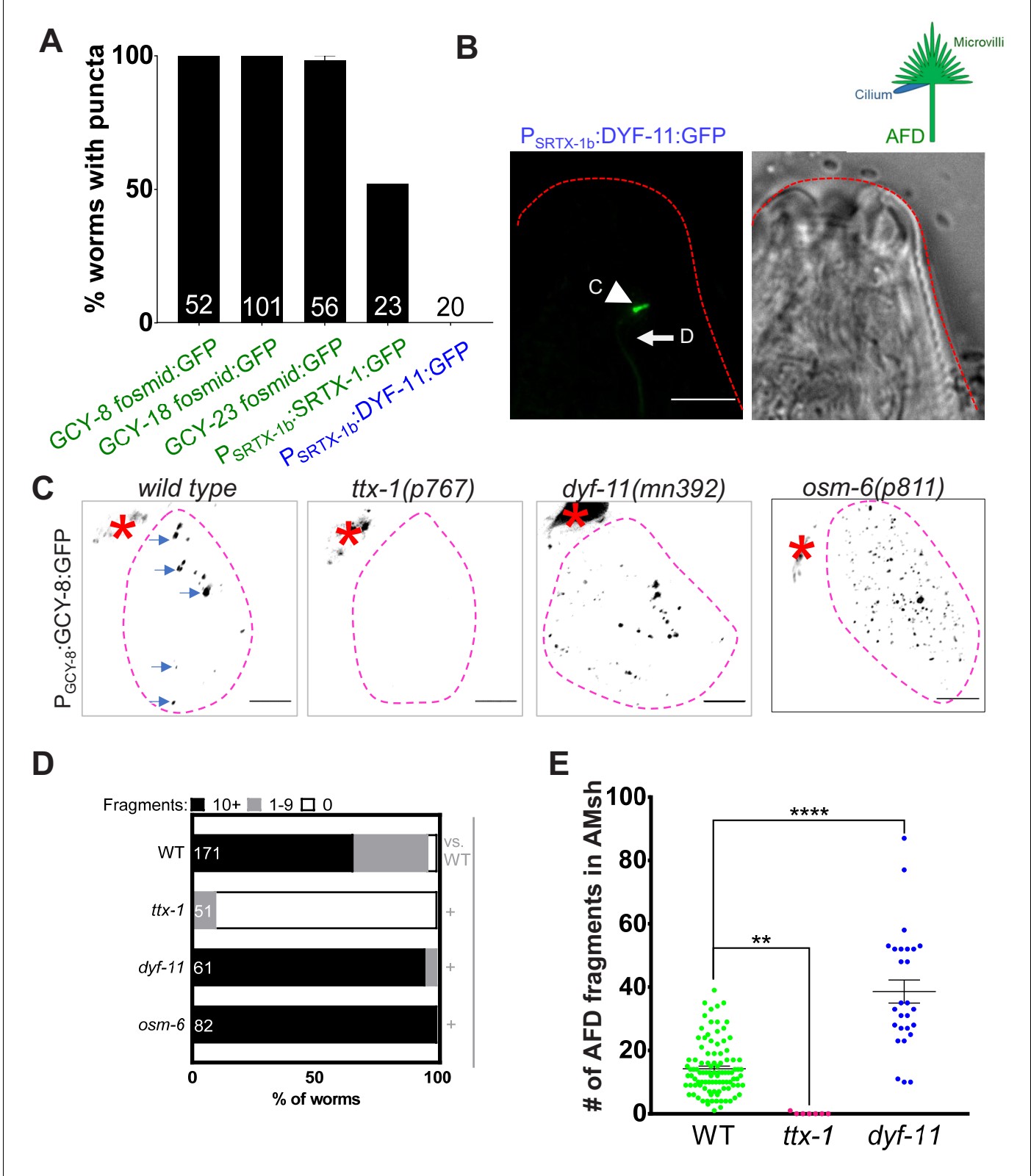

**Figure 3.** AMsh glia engulf AFD–NRE microvilli but not cilia. (**A**) AFD–NRE-labeled fragments observed in different transgenic animal strains. Each strain has a different tagged fusion protein, driven by a different AFD-specific promoter, localizing to either microvilli (green) or cilium (blue). X-axis: genotype; Y-axis: percent animals with AFD–NRE-labeled puncta inside AMsh soma. N: number of animals analyzed. (**B**) Schematic depicting the two compartments of the AFD–NRE, which is an array of ~45 actin-based microvilli (green) and a single microtubule-based cilium (blue). Fluorescence and

*Figure 3 continued on next page*

Figure 3 continued

DIC micrographs showing expression of ciliary DYF-11:GFP, under an AFD neuron-specific promoter, in AFD cilia. C: cilia (arrowhead); D: AFD dendrite (arrow). (C) Fluorescence micrograph panel showing AFD–NRE tagged puncta (blue arrows) within AMsh glial cell soma (magenta outline) in different genetic backgrounds as noted. AFD cell body (red asterisk). Scale bar: 5 μm. (D) Population counts of animals with AMsh glial puncta. Refer *Figure 2C* for data presentation details. Alleles used: *ttx-1(p767)*, *dyf-11(mn392)*, and *osm-6(p811)*. (+) p<0.05 compared to wild type, (–) p≥0.05 compared to wild type. (E) Median puncta counts and N (number of animals): *wild type* (14 ± 1 puncta, n = 78 animals), *ttx-1(p767)* (0.1 ± 0.1 puncta, n = 7 animals), and *dyf-11(mn392)* (38.6 ± 3.6 puncta, n = 27 animals). Refer *Figure 2D* for data presentation details. NRE: neuron-receptive ending.

The online version of this article includes the following source data for figure 3:

**Source data 1.** AMsh glia engulf AFD–NRE microvilli but not cilia.

CED-6/GULP and CED-7/ABCA1, which function with CED-1/MEGF10 in *C. elegans* apoptotic phagocytosis and in other species (*Flannagan et al., 2012*; *Hamon et al., 2006*; *Morizawa et al., 2017*; *Reddien and Horvitz, 2004*; *Zhou et al., 2001*), does not block engulfment either (*Figure 4C*). Further, mutations in tyrosine kinases related to MeRTK, required for astroglial engulfment of neuronal debris in vertebrates (*Chung et al., 2013*), also seem to not be required for AMsh engulfment of AFD–NRE (*Figure 4—figure supplement 1A*; *Popovici et al., 1999*).

Loss of the conserved phosphatidylserine receptor PSR-1/PSR has defects in apoptotic cell corpse engulfment in *C. elegans* and zebrafish (*Hong et al., 2004*; *Wang et al., 2003*). Remarkably, deletion of *psr-1* dramatically reduces AFD–NRE engulfment by AMsh glia (*Figure 4D, E*). Expression of the PSR-1C long isoform in AMsh glia rescues *psr-1* mutant defects significantly (*Figure 4D*), suggesting that PSR-1 acts in glia to promote NRE uptake. Consistent with this function, a GFP:PSR-1 translational reporter expressed under an AMsh-glia-specific promoter localizes to glial membranes, including those around AFD–NRE microvilli (*Figure 4F*, F').

If PSR-1 recognizes PS on AFD–NRE membranes to mediate engulfment, we reasoned it should act downstream of TAT-1. We therefore constructed and analyzed *psr-1; tat-1* double mutants. Unlike *tat-1* single mutants that show increased NRE engulfment, *psr-1; tat-1* animals exhibit reduced engulfment similar to *psr-1* single mutants (*Figure 4D*). Thus, PSR-1 acts downstream of TAT-1.

The transthyretin protein TTR-52 mediates binding between PS and PSR-1 (*Neumann et al., 2015*; *Wang et al., 2010*). Supporting the PSR-1 results, we found that a mutation in *ttr-52* also reduces NRE uptake to a similar extent as mutations in *psr-1* (*Figure 4D*). In addition, we found that *psr-1; ttr-52* double mutants show no significant enhancement of puncta defects compared to either single mutant, suggesting that PSR-1 and TTR-52 function within the same pathway for PS recognition by AMsh glia (*Figure 4D*).

## Integrin α-subunit PAT-2 regulates glial engulfment with PSR-1

Although *psr-1* loss reduces puncta numbers (and by inference, NRE engulfment) dramatically, we noted that neuronal fragment uptake is not completely eliminated (*Figure 4D*). This suggested that another receptor may be involved. Integrins function with MeRTK to promote photoreceptor cell outer segment engulfment by retinal RPE glia (*Mao and Finnemann, 2012*), and the *C. elegans* genome encodes two α-integrin subunits, INA-1 and PAT-2, both of which are implicated in apoptotic cell phagocytosis in *C. elegans* (*Hsieh et al., 2012*; *Neukomm et al., 2014*; *Sáenz-Narciso et al., 2016*). We found that while a mutation in *ina-1* has no effect on NRE engulfment (*Figure 4—figure supplement 1A*), loss of PAT-2 by RNA interference (RNAi) significantly blocks AFD–NRE phagocytosis (*Figure 4G*). Further, *pat-2* RNAi strongly enhances glia engulfment defects of *psr-1* mutants (*Figure 4G*). Thus, PAT-2/α-integrin and PSR-1 appear to act together for glial engulfment of AFD–NRE.

Curiously, not only do mutations in *ced-1* not block the appearance of puncta in glia, we found that *ced-1(e1754)* strong loss-of-function mutant animals actually exhibit enhanced puncta numbers compared to wild-type animals (*Figure 4C*). We found that *pat-2* RNAi did not block this enhanced engulfment defect of *ced-1(e1754)* animals (*Figure 4—figure supplement 1B*), suggesting that PAT-2 and CED-1 likely do not function synergistically as PS-receptors for glia-dependent phagocytosis. In line with this, while *psr-1 ced-1* double mutant animals exhibit a slightly higher fraction of animals with no puncta, *ced-1* in fact suppresses the synergistic engulfment defects seen in *psr-1; pat-2* (RNAi) animals (*Figure 4—figure supplement 1B*). This suggests that either *ced-1* has a minor role

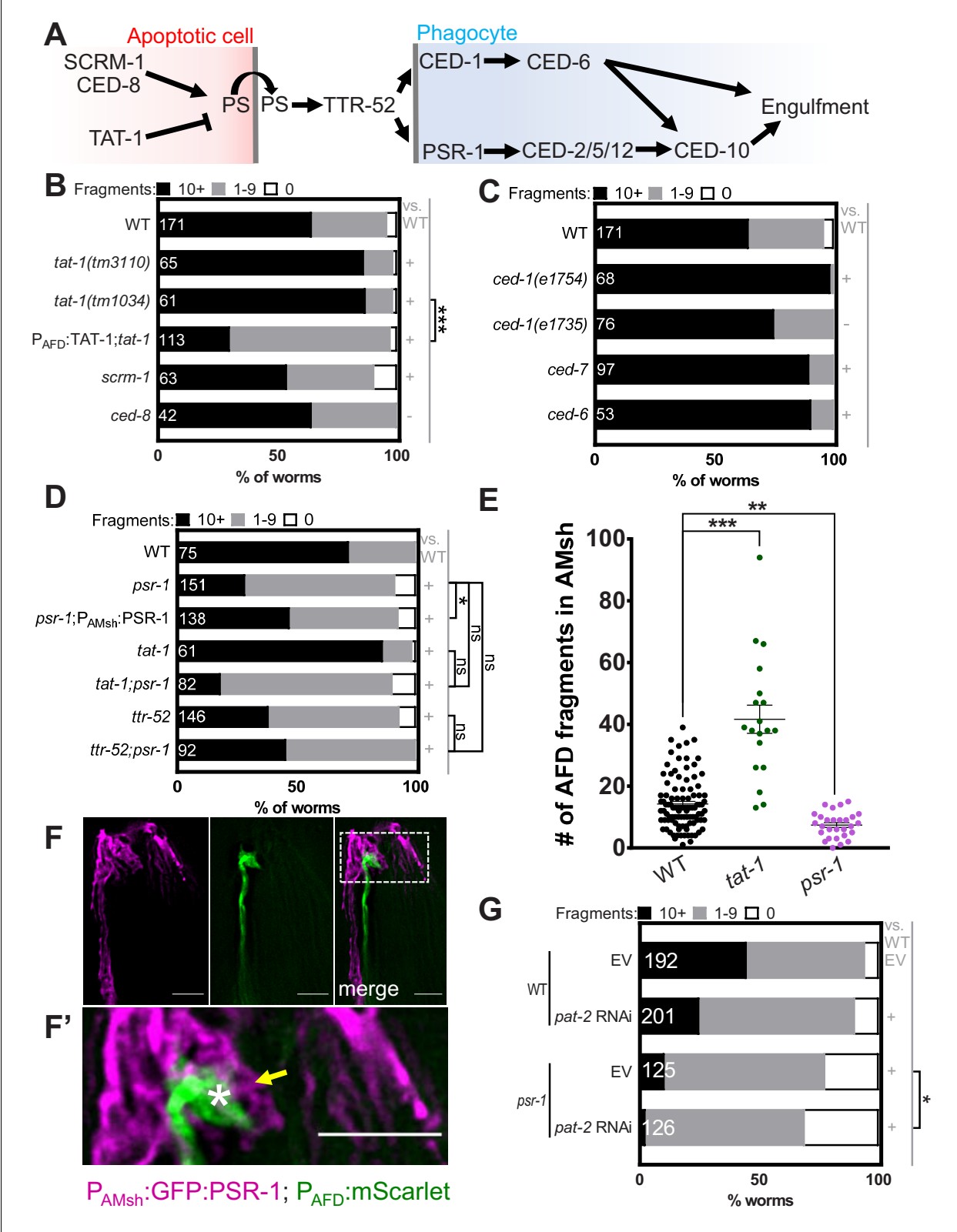

**Figure 4.** Engulfment of AFD–NRE by AMsh glia requires the phosphatidylserine receptor PSR-1 and integrin PAT-2. (**A**) Schematic of the genetic pathway underlying apoptotic corpse engulfment in *C. elegans*. (**B–D**) Population counts of animals with AMsh glia puncta. Refer *Figure 2C* for data presentation details. (+) p<0.05 compared to wild type, (−) p≥0.05 compared to wild type. (**B**) Alleles used in this graph: *tat-1(tm3110)*, *tat-1(tm1034)*, *scrm-1(tm805)*, and *ced-8(n1819)*. (**C**) Alleles used in this graph: *ced-1(e1754)*, *ced-1(e1735)*, *ced-7(n2094)*, and *ced-6(n1813)*. (**D**) Alleles used in this

Figure 4 continued

graph: *psr-1(tm469)*, *tat-1(tm1034)*, and *ttr-52(tm2078)*. (E) Quantification of puncta within AMsh cell soma in listed mutants. Refer *Figure 2D* for data presentation details. Median puncta counts and N (number of animals): *wild type* (14 ± 1 puncta, n = 78 animals), *psr-1(tm469)* (7.4 ± 0.8 puncta, n = 28 animals), and *tat-1* (41.6 ± 4.6 puncta, n = 19 animals). (F) Fluorescence micrograph of a transgenic animal with GFP tagged PSR-1 expressed specifically in AMsh glia (magenta) localizing on the apical membrane around AFD–NRE (green). GFP:PSR localizes to apical membrane in AMsh glia (yellow arrow) around AFD–NRE (asterisk). Scale bar: 5 µm. (F') Zoom of box in two-color merged image. (G) RNAi (control *pat-2*) in wild-type or *psr-1(tm469)* mutant animals. Refer *Figure 2C* for data presentation details. EV: empty vector control. NRE: neuron-receptive ending.

The online version of this article includes the following source data and figure supplement(s) for figure 4:

**Source data 1.** Engulfment of AFD–NRE by AMsh glia requires the phosphatidylserine receptor PSR-1 and integrin PAT-2.
**Figure supplement 1.** Engulfment of AFD–NRE by AMsh glia does not depend on some RTK or CED-1/MEGF10/Draper.

in engulfment as a PS-receptor or its role in this glia-dependent phagocytosis is non-canonical. To examine this further, we also asked if *ttr-52* acts with *ced-1*. The *ced-1;ttr-52* double mutant had the same increased glia puncta as *ced-1* single mutants, suggesting that *ced-1* acts genetically downstream of *ttr-52* (*Figure 4—figure supplement 1C*). Finally, the *ced-1; ttr-52; psr-1* triple mutant also phenocopied *ced-1* single mutants in having increased number of glia puncta, suggesting again that CED-1 acts downstream of PSR-1 and TTR-52. These data raise the possibility that in NRE engulfment CED-1 may instead act in phagolysosome maturation downstream of PS recognition, as has been observed for CED-1 in other contexts (*Yu et al., 2006*).

## The CED-2/5/12 ternary GEF complex acts in AMsh glia to promote engulfment

The ternary complex of CED-2/CrkII, CED-5/DOCK1, and CED-12/ELMO1 acts downstream of PSR-1 for apoptotic cell engulfment (*Reddien and Horvitz, 2004*; *Wang et al., 2003*). We found that animals bearing mutations in *ced-2, ced-5*, or *ced-12* exhibit reduced AFD–NRE puncta in AMsh glia (*Figure 5A*). Furthermore, expression of the CED-12B isoform in AMsh glia is sufficient to rescue *ced-12* mutant defects (*Figure 5A*). We conclude, therefore, that the CED-2/CED-5/CED-12 complex also likely regulates engulfment of AFD–NREs.

## Glial Rac1 GTPase CED-10 controls rate of engulfment

CED-2/CED-5/CED-12 act as a GEF for the Rac1 GTPase CED-10, a major downstream effector of a number of apoptotic phagocytosis pathways (*Flannagan et al., 2012*; *Reddien and Horvitz, 2004*; *Wang and Yang, 2016*; *Figure 4A*). CED-10 is also implicated in engulfment of photoreceptor outer segments by RPE glia-like cells in mammals and debris of injured axons by glia in *Drosophila* (*Kevany and Palczewski, 2010*; *Lu et al., 2014*; *Nichols et al., 2016*). We found that two loss-of-function mutations in *ced-10*, or overexpression of dominant-negative CED-10$^{T17N}$, block nearly all engulfment of AFD–NRE fragments by AMsh glia (*Figure 5B–D*). Specifically, in two different alleles, very few puncta are observed in glia (*ced-10(n3246)* [3.08 ± 0.79, n = 39] and *ced-10(n1993)* [2.4 ± 0.6 puncta, n = 24 animals] vs. *wild type* [14 ± 1 puncta, n = 78 animals]). Furthermore, barely any mutant animal had >10 puncta (*ced-10(n3246)* = 0.81%, n = 124; and *ced10(n1993)* = 2.78%, n = 72; compared to *wild type* = 64%, n = 171). Expressing CED-10 only in AMsh glia completely restores engulfment to *ced-10* loss-of-function mutants (*Figure 5B–D*).

To determine how CED-10 functions with respect to CED-2/CED-5/CED-12 and PSR-1, we generated *psr-1; ced-10* and *ced-12; ced-10* double mutants. Both strains show strong defects in puncta numbers reminiscent of *ced-10* single mutants (*Figure 5E*). Furthermore, transgenic expression of CED-10 is sufficient to overcome the partial loss of NRE engulfment in *psr-1* mutants (*Figure 5E*). Our data are consistent with the interpretation that, like in cell corpse engulfment, CED-10/Rac1 GTPase likely functions in glia downstream of CED-2/CED-5/CED-12 and PSR-1, to promote AMsh glial engulfment of NREs. This activation is specific as mutations in another CED-10 activator, UNC-73/TRIO, do not affect NRE uptake (*Figure 4—figure supplement 1A*; *Lundquist et al., 2001*; *Sáenz-Narciso et al., 2016*).

Unexpectedly, expression of constitutive active CED-10$^{G12V}$ also results in reduced engulfed puncta (*Figure 5D*). This may indicate that a GTPase cycle is needed for engulfment to proceed (*Bernards and Settleman, 2004*; *Sáenz-Narciso et al., 2016*; *Singhvi et al., 2011*; *Takai et al.,*

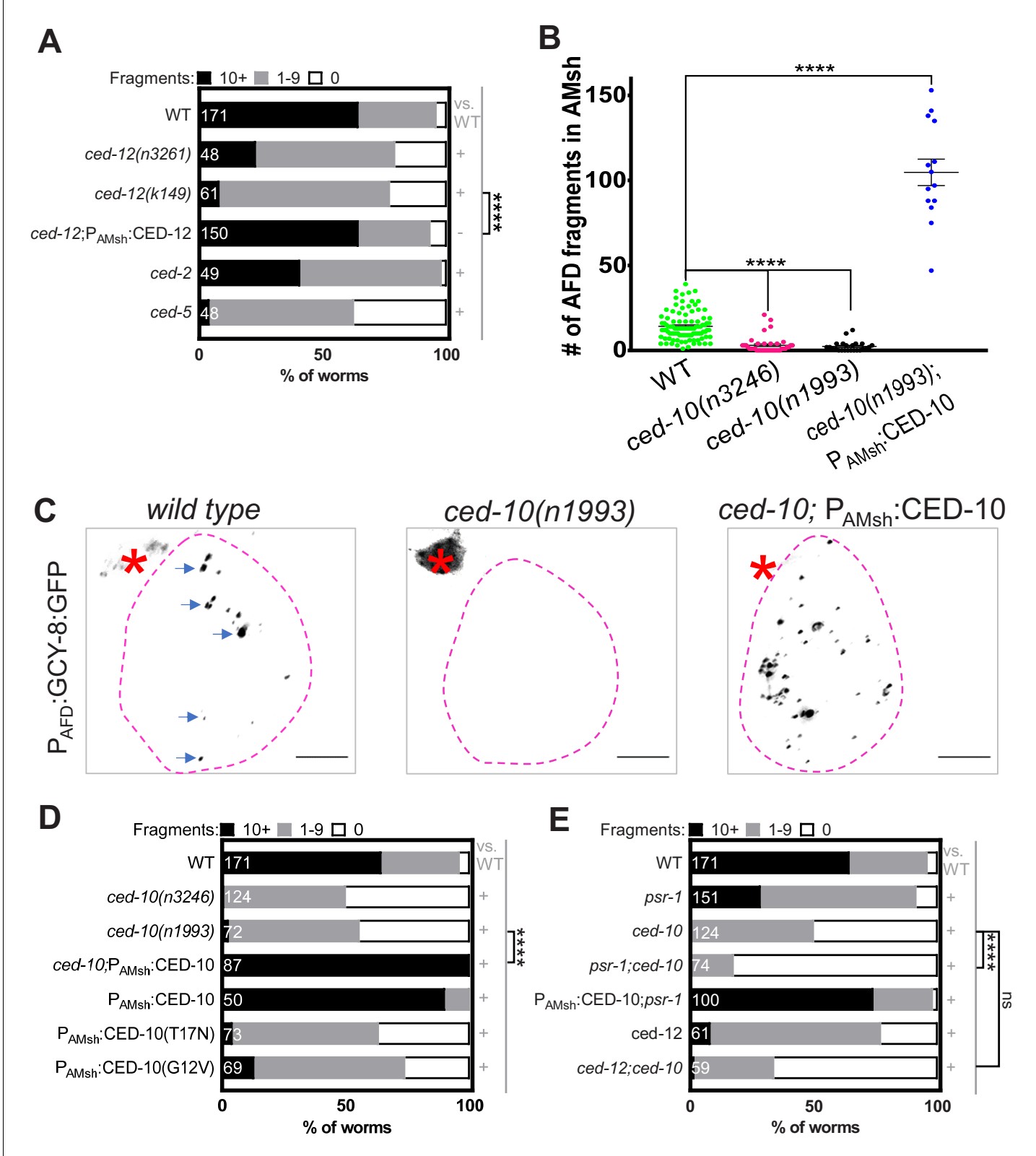

**Figure 5.** Phagocytosis pathway components, glial CED-10 levels, and actin remodeling actively control rate of engulfment. (**A**) Population counts of animals with AMsh glial puncta in the indicated genetic backgrounds. Refer *Figure 2C* for data presentation details. (+) p<0.05 compared to wild type, (–) p≥0.05 compared to wild type. Alleles used in this graph: *ced-12(n3261)*, *ced-12(k149)*, *ced-2(e1752)*, and *ced-5*(n1812). (**B**) Quantification of puncta within AMsh cell soma in phagocytosis pathway mutants. Refer *Figure 2D* for data presentation details. Median puncta counts and N (number of

*Figure 5 continued on next page*

Figure 5 continued

animals): *wild type* (14 ± 1 puncta, n = 78 animals), *ced-10(n1993)* (2.4 ± 0.6 puncta, n = 24 animals), *ced-10(n3246)* (3.08 ± 0.79, n = 39), and P$_{AMsh}$:CED-10 (104.7 ± 7.8 puncta, n = 14 animals). (C) Panel showing AFD–NRE tagged puncta (blue arrows) within AMsh glial cell soma (magenta outline) in different genetic backgrounds as noted. AFD cell body (red asterisk). Scale bar: 5 µm. (D, E) Population counts of animals with AMsh glial puncta in genetic backgrounds indicated. Refer **Figure 2C** for data presentation details. (+) p<0.05 compared to wild type, (–) p≥0.05 compared to wild type. (D) Alleles used in this graph: *ced-10(n3246)* and *ced-10(n1993)*. CED-10$^{G12V}$ and CED-10$^{T17N}$ is a constitutively active or dominant negative form of CED-10, respectively. (E) Alleles used in this graph: *psr-1(tm469)*, *ced-10(n3246)*, and *ced-12 (k149)*. NRE: neuron-receptive ending.

The online version of this article includes the following source data and figure supplement(s) for figure 5:

**Source data 1.** Phagocytosis pathway components, glial CED-10 levels, and actin remodeling actively control rate of engulfment.
**Figure supplement 1.** The actin regulator WSP-1 can regulate engulfment cell-autonomously in AMsh glia.

2001; **Teuliere et al., 2014**). Alternatively, it may be that this form of the protein promotes hyperefficient engulfment, which does not leave much NRE to be engulfed. Supporting the latter model, the AFD–NRE is significantly shorter in CED-10$^{G12V}$ mutants (see below). Furthermore, overexpression of wild-type CED-10, but not of wild-type PSR-1 or CED-12, increases NRE engulfment (**Figure 4D**, **Figure 5A,D**). Glial CED-10 is, therefore, both necessary and sufficient to regulate the rate at which AMsh glial engulf AFD–NRE fragments.

During apoptotic cell engulfment, CED-10 executes phagocytic arm extension by mediating actin remodeling (**Wang and Yang, 2016**). We, therefore, examined animals bearing a loss-of-function mutation in *wsp-1*, which encodes an actin polymerization factor, and found a block in NRE engulfment (**Figure 5—figure supplement 1A**). As with overexpression of CED-10, increasing levels of WSP-1 specifically in AMsh glia also lead to increased NRE engulfment (**Figure 5—figure supplement 1A**). These results suggest that CED-10-dependent actin remodeling is the rate-limiting step for the engulfment of AFD–NREs by glia.

## Glial engulfment tracks neuron activity post development

Previous studies showed that cyclic-nucleotide-gated (CNG) ion channels localize to the AFD cilium base and are required for AFD neuron firing in response to temperature stimuli (**Cho et al., 2004**; **Ramot et al., 2008**; **Satterlee et al., 2004**). These channels are mis-localized in cilia-defective mutants (**Nguyen et al., 2014**). Independently, it has been shown that cilia-defective mutants exhibit deficits in thermotaxis behavior (**Tan et al., 2007**). Since we found that cilia-defective mutants have increased engulfment (**Figure 3**), these taken together prompted us to examine the role for neuron activity in glial engulfment directly.

We examined animals defective in TAX-2, the sole CNG β-subunit in the *C. elegans* genome, or in TAX-4 and CNG-3, α-subunits that function together in AFD (**Cho et al., 2004**; **Hellman and Shen, 2011**; **Satterlee et al., 2004**) for engulfment defects. Glia in mutant animals accumulate extra puncta (*tax-2*: 28.1 ± 2 puncta, n = 37; *tax-4; cng-3* double mutants: 23.8 ± 2.3 puncta, n = 17) (**Figure 6A–C**), and in *tax-2* mutants, a larger fraction of the animal population has >10 puncta (*tax-2*, 99%, n = 92 animals; *wild type*, 65%, n = 171 animals) (**Figure 6C**). Conversely, we assessed the consequence of increasing the levels of cGMP, which promotes CNG channel opening, by mutating the cGMP degrading enzymes PDE-1 and PDE-5 expressed in AFD neurons (**Ramot et al., 2008**; **Singhvi et al., 2016**). We found that *pde-1; pde-5* double mutant animals have reduced glia puncta numbers compared to wild type (7.1 ± 1.4, n = 11 vs. 14 ± 1, n = 78) (**Figure 6B, C**). Finally, acute and cell-specific chemogenetic silencing of AFD using a histamine-gated chloride channel (**Pokala et al., 2014**) expressed under an AFD-specific promoter leads to puncta enrichment in AMsh glia within 24 hr (**Figure 6E, F**). Thus, AFD activity levels reciprocally affect AFD–NRE engulfment levels and can do so acutely.

Accumulation of glial puncta in AFD activity mutants could result from increased engulfment rates or, alternatively, from decreased puncta degradation. We favor the former model as we found that the increase in puncta number seen in *tax-2* mutant glia is entirely suppressed by loss of CED-10 (**Figure 6B, C**). Likewise, we also observed significant suppression in *tax-2; psr-1(tm469)* double mutants compared to *tax-2* alone; and this suppression is enhanced further by *pat-2* (RNAi) (**Figure 6C, D**). Loss of *ced-10* also suppresses excess engulfment following acute chemogenetic silencing of AFD (**Figure 6F**). Our findings are therefore consistent with neuron activity controlling NRE engulfment through the CED-10 pathway.

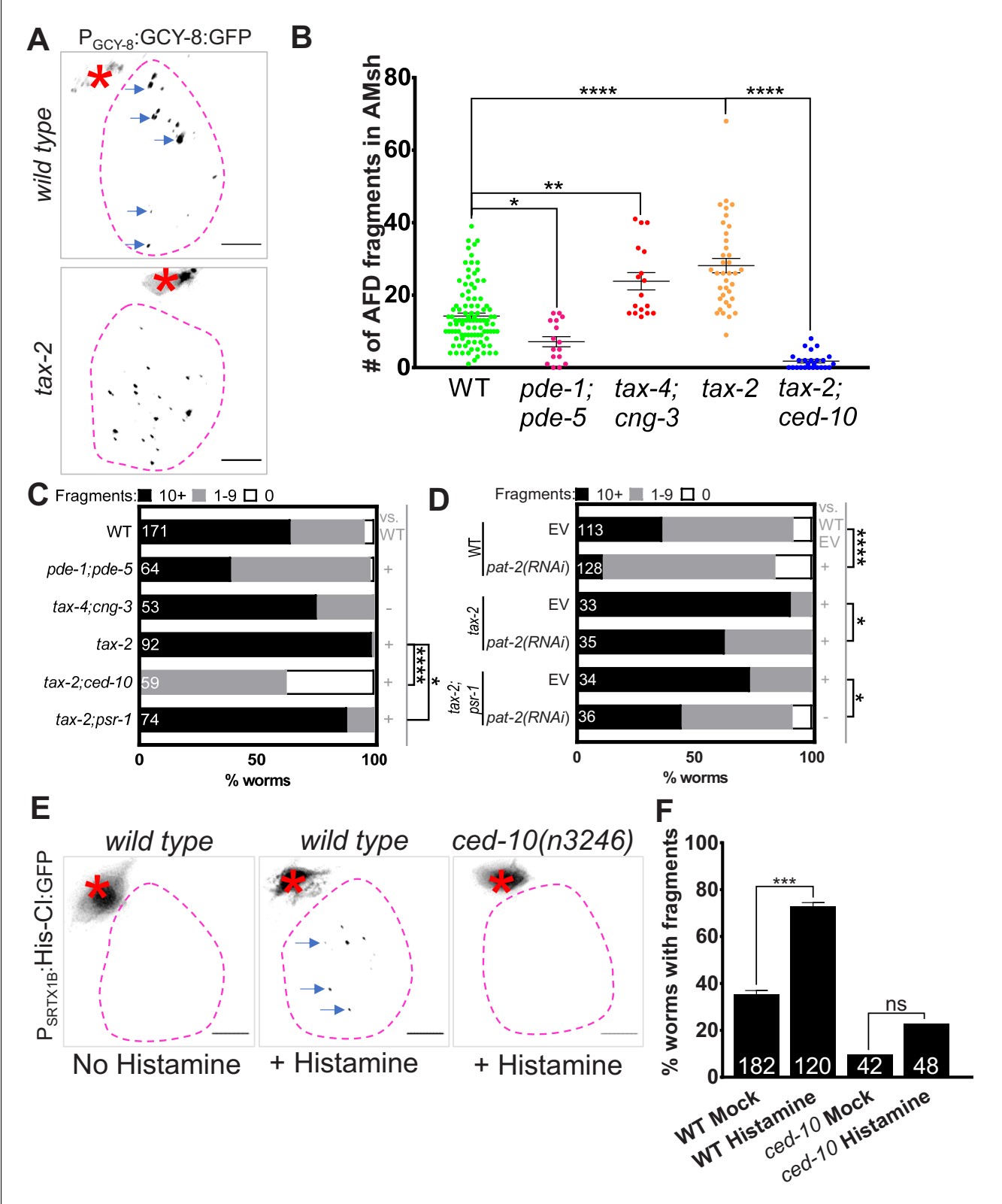

**Figure 6.** Glial phagocytic pathway tracks neuron activity to regulate AFD–NRE engulfment rate. (**A**) Panel showing AFD–NRE tagged puncta (blue arrows) within AMsh glial cell soma (magenta outline) in different genetic backgrounds, as noted. AFD cell body (red asterisk). Scale bar: 5 μm. (**B**) Quantification of puncta within AMsh cell soma in phagocytosis pathway mutants. Refer *Figure 2D* for data presentation details. Median puncta counts and N (number of animals): *wild type* (14 ± 1 puncta, n = 78 animals), *pde-1(nj57) pde-5(nj49)* double mutant animals (7.1 ± 1.4, n = 11 animals), *tax-4*

*Figure 6 continued on next page*

*Figure 6 continued*

(*p678);cng-3(jh113)* double mutants (23.8 ± 2.4 puncta, n = 17 animals), *tax-2(p691)* (28.1 ± 2 puncta, n = 37 animals), and *ced-10(n3246); tax-2(p691)* double mutants (1.8 ± 0.5 puncta, n = 25 animals). (C, D) Population counts of animals with AMsh glial puncta in genetic backgrounds indicated. Refer **Figure 2C** for data presentation details. (+) p<0.05 compared to wild type, (–) p≥0.05 compared to wild type. (C) Alleles used in this graph: *pde-1(nj57), pde-5(nj49), tax-4(p678), cng-3(jh113), tax-2(p691), ced-10(n3246),* and *psr-1(tm469).* (D) Alleles used in this graph: *tax-2(p691),* and *psr-1(tm469).* EV: empty vector control. (E) Percent *wild type* or *ced-10(n3246)* mutant animals with observable GFP+ puncta with or without histamine. N: number of animals. (F) Quantification of percent animals with puncta in AMsh glia (Y-axis) in transgenic strains carrying a histamine-gated chloride channel, with/out histamine activation as noted (X-axis). NRE: neuron-receptive ending.
The online version of this article includes the following source data for figure 6:

**Source data 1.** Glial phagocytic pathway tracks neuron activity to regulate AFD–NRE engulfment rate.

## Glial engulfment regulates AFD–NRE shape and thermotaxis behavior

What might be the function of AFD–NRE engulfment by glia? To test this, we examined AFD–NRE shape by 3D super-resolution imaging of transgenic mutants bearing a tagged reporter that specifically marks AFD–NRE microvilli. We found that *ced-10* loss of function, or AMsh glia-specific overexpression of dominant negative CED-10$^{T17N}$, results in elongated AFD–NRE microvilli (**Figure 7A, B**, **Figure 7—figure supplement 1B**). By contrast, overexpressing wild-type CED-10, which has excess puncta, produces shorter AFD–NRE microvilli, and this defect worsens with age (**Figure 7A, B**). Furthermore, overexpressing GTP-locked CED10$^{G12V}$ also leads to shorter AFD–NRE microvilli even though it paradoxically has reduced number of puncta in glia (**Figure 7—figure supplement 1A, B**) consistent with the idea that engulfment in this strain may be so efficient that no NREs remain to be engulfed. Thus, AMsh glial engulfment of NRE fragments is important for regulating the AFD–NRE microvilli length.

When placed on a temperature gradient, *C. elegans* seek their temperature of cultivation, $T_c$ (**Hedgecock and Russell, 1975**; **Figure 7C–F**, wild-type data in black line). This animal behavior depends on thermosensory transduction at the AFD–NRE (**Goodman and Sengupta, 2018**; **Mori and Ohshima, 1995**). Previous studies have shown that animals with defects in AFD–NRE shape also exhibit defects in this thermosensory behavior. Consistent with this, we found that *ced-10* mutants exhibit altered thermosensory behavior. While wild-type animals reared at 25°C migrate to their $T_c$ = 25°C on a linear temperature gradient, *ced-10* mutants prefer cooler temperatures (**Figure 7C, D**). Furthermore, animals carrying integrated transgenes overexpressing CED-10 only in AMsh glia also exhibit athermotactic defects regardless of the cultivation temperatures (**Figure 7E, F**). We conclude, therefore, that AFD–NRE engulfment by AMsh glia is required for appropriate animal thermotaxis behaviors.

The behavior defects we observed are consistent with the thesis that reduced neuron activity drives glial engulfment. The athermotactic behavior of CED-10 overexpression strains mimics similar defects of *tax-2* or *tax-2; tax-4* double mutant animals, and both manipulations lead to increased puncta and reduced neuron activity (**Figure 7—figure supplement 1C, D**; **Cho et al., 2004**; **Satterlee et al., 2004**). Likewise, the cryophilic behavior of *ced-10* mutants, which have reduced glia puncta, is similar to that observed in other mutants with increased AFD cGMP levels (**Singhvi et al., 2016**). We favor the model that activity-dependent glial engulfment of NRE is one mechanism by which AMsh glia and AFD coordinate regulation of NRE shape and animal thermosensory behavior.

## Discussion

We report our discovery that *C. elegans* glia, like glia of other species, engulf associated neuron endings, highlighting evolutionary conservation of this critical glial function (**Figure 8**). Exploiting unique features of our experimental model, we demonstrate that glial CED-10 levels dictate engulfment rates, revealing that glia drive neuronal remodeling and do not just passively clear shed neuronal debris. Indeed, we demonstrate that engulfment is required for post-developmental maintenance of sensory NRE shape and behavior. This also extends a role for glial engulfment in the active sensory perception of temperature. Importantly, our studies allow us to directly demonstrate at single-cell resolution that pruning of individual neurons by a single glia modifies animal behavior. This, in conjunction with our finding that phagocytosis is impacted by neuronal activity states, demonstrates important physiological relevance.

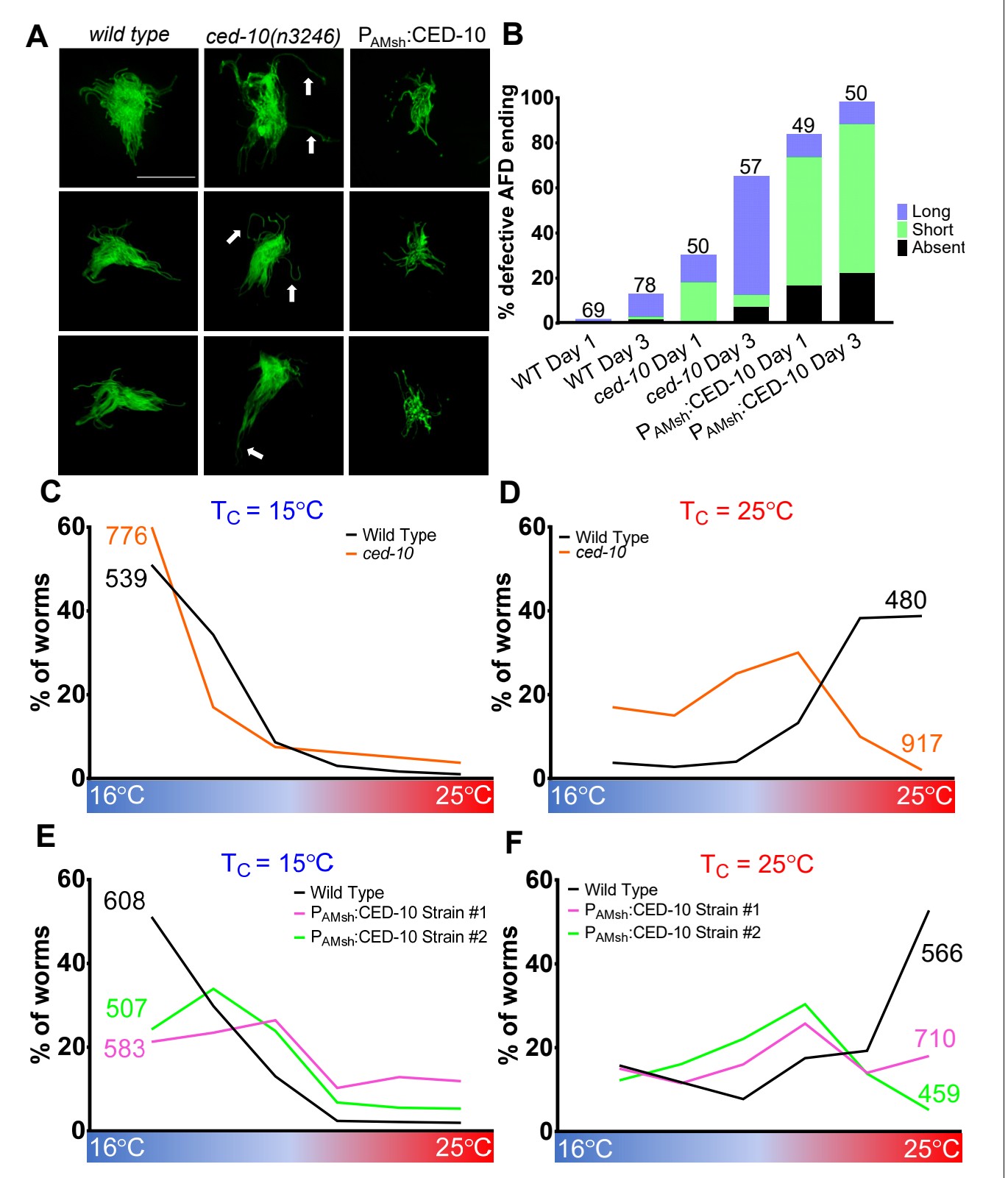

**Figure 7.** AMsh glial engulfment of AFD–NRE modulates AFD–NRE shape and animal thermosensory behavior. (**A**) AFD–NRE microvilli labeled with GFP in day 3 adult animals of genotypes as indicated. Three representative images are shown for each genotype. Scale bar: 5 µm (**B**) Quantification of percent animals with defective AFD–NRE microvilli shape. N: number of animals scored. (**C–F**) Thermotaxis behavior assays for animals of indicated genotype raised at 15°C (**C, E**) or 25°C (**D, F**). Animals assayed 24 hr post-mid-L4 larval stage. N: number of animals. NRE: neuron-receptive ending.
*Figure 7 continued on next page*

*Figure 7 continued*

The online version of this article includes the following source data and figure supplement(s) for figure 7:

**Source data 1.** AMsh glial engulfment of AFD–NRE modulates AFD–NRE shape and animal thermosensory behavior.
**Figure supplement 1.** AMsh glial CED-10 tracks neuron activity to regulate AFD–NRE engulfment.

## Controlled tuning of the phagocytosis machinery

Our studies reveal a fundamental distinction between glia-dependent phagocytosis and other modes of engulfment. Apoptotic cell phagocytosis, glial clearance of injury-induced neuronal debris, and related engulfment events are all-or-none phenomena: engulfment either occurs or does not. By contrast, we show here that in AMsh glia engulfment rate is dynamically tuned throughout animal life to modulate NRE morphology, impacting animal behavior. The molecular parallels between the engulfment machinery in the peripheral sense-organ AMsh glia and other CNS glial engulfment lead us to posit that controlled phagocytosis may similarly regulate glial engulfment in other settings.

## Distinct receptors mediate PS-dependent glial pruning

Accompanying this more versatile engulfment program is a shift in the relevance of specific engulfment receptors. Apoptotic phagocytosis in *C. elegans* relies predominantly on CED-1, with the PS-receptor PSR-1 playing a minor role (*Wang et al., 2003*; *Wang and Yang, 2016*). Surprisingly, while CED-1 is dispensable for pruning by AMsh glia, we identified PSR-1/PS-receptor as a novel regulator of glial pruning. Why do CED-1 and PSR-1 have differing valence in apoptotic phagocytosis and glial pruning? One possibility is that this difference in receptors reflects the size of particles engulfed. Supporting this notion, engulfment of small cell process debris of the *C. elegans* tail-spike cell is also independent of CED-1 (*Ghose et al., 2018*).

We identified PSR-1 and integrins as a PS-receptor driving AMsh glial engulfment of AFD–NRE. Other PS-receptors that have been shown to regulate glial engulfment across species include CED-1/MEGF10/Draper, MerTK, and GPR56, and it is likely that yet others await identification

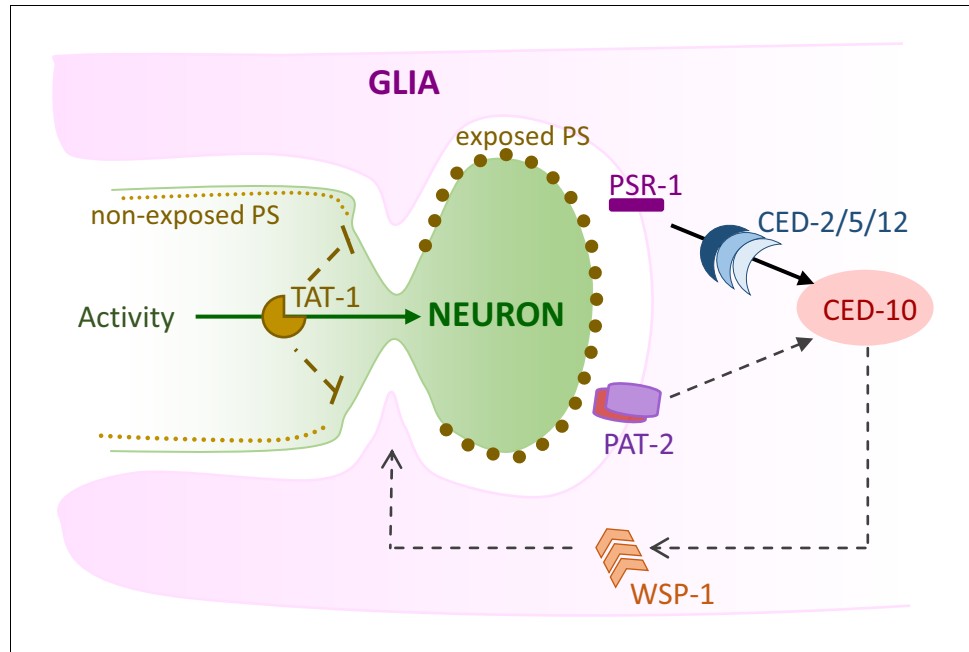

**Figure 8.** Model of AMsh glial engulfment of AFD–NRE. Model depicting molecular machinery driving engulfment of AFD neuron microvilli by AMsh glia. TAT-1 maintains phosphatidylserine on the inner plasma leaflet. Neuron activity negatively regulates engulfment. The phosphatidylserine receptor PSR-1 signals via ternary GEF complex CED-2/5/12 to activate Rac1 GTPase CED-10, along with PAT-2/integrin. CED-10 and its downstream effector, WSP-1, drive engulfment of AFD neuron microvilli fragments. NRE: neuron-receptive ending.

(*Chung et al., 2013*; *Freeman, 2015*; *Hilu-Dadia and Kurant, 2020*; *Kevany and Palczewski, 2010*; *Li et al., 2020*; *Nomura-Komoike et al., 2020*; *Raiders et al., 2021*; *Tasdemir-Yilmaz and Freeman, 2014*; *Vecino et al., 2016*). This then raises the question of why one analogous glial function of pruning would require different receptors. We speculate that this may reflect the molecular heterogeneity across glia and/or the context of engulfment (*Raiders et al., 2021*).

## Mediators of PS exposure in *C. elegans* glial pruning

PS exposure has emerged as a classic engulfment signal for both apoptotic phagocytosis and glial pruning, but how this is regulated remains enigmatic. We identify this as a conserved feature in *C. elegans* glial engulfment and implicate the phospholipid transporter TAT-1/ATP8A in this process. TAT-1 is a member of the type 4 family P4 ATPases, which flip PS from exoplasmic to cytoplasmic membrane leaflets (*Andersen et al., 2016*). We note that murine P4-ATPases ATP8A1 and ATP8A2 are expressed in the nervous system, and knockout mice exhibit deficient hippocampal learning, sensory deficits, cerebellar ataxia, mental retardation, and spinal cord degeneration, and shortened photoreceptor NRE length (*Coleman et al., 2014*). Given this intriguing parallel, it will be interesting to probe whether ATP8A similarly modulates glial pruning in mammals.

We also identify the PS bridging molecule TTR-52 as a regulator of pruning. It is also implicated in apoptotic phagocytosis and nerve regeneration (*Neumann et al., 2015*; *Wang et al., 2010*). Retinal RPE glia and cortical astrocytes also require PS-bridging opsonins (Gas6 and MFGE8) to engulf neuron fragments (*Bellesi et al., 2017*; *Kevany and Palczewski, 2010*).

Whether all glia require PS opsonization for pruning remains to be determined.

## Glia direct pruning with subcellular precision

Our finding that proper animal behavior requires precise levels of NRE engulfment by glia suggests that engulfment must proceed with extraordinary specificity so that behavior is optimal. Indeed, we find that AMsh glia prune AFD–NRE with subcellular precision. While AFD's actin-rich microvilli are removed by glia, its adjacent microtubule-based cilium is not. Aberrantly excessive/reduced pruning correlate with disease in mammals, hinting that similar subcellular precision in marking fragments/endings for engulfment might be involved (*Chung et al., 2015*; *Wilton et al., 2019*). How this precision is regulated will be fascinating to explore.

## Peripheral sense-organ glia pruning modulates NRE shape and animal sensory behaviors

A role for pruning in normal neural functions has so far been investigated for central nervous system glia (astrocytes, microglia, retinal glia). Peripheral glia of the inner ear are known to activate phagocytosis only in injury settings (*Bird et al., 2010*). Our studies demonstrate that pruning of sensory neuron endings by glia is required for accurate sensory perception. Thus, glial pruning is conserved in both CNS and PNS and is executed for normal neural functions by analogous molecular mechanisms. While these studies identify glial pruning as a mechanism to control NRE shape in response to activity states, we note that it is likely that AMsh glia and AFD neuron cooperate through multiple mechanisms to regulate AFD–NRE shape and animal thermosensory behaviors, including some that we previously identified (*Singhvi et al., 2016*; *Wallace et al., 2016*). Such regulatory complexity might reflect the fact that appropriate thermosensory behaviors are critical for animal survival.

## Active pruning versus passive clearance of debris

An outstanding question in understanding the role of glia is whether glia actively prune NREs and neuron fragments or passively clear shed debris. Three lines of evidence in this study lead us to conclude that AMsh glia actively drive engulfment rather than passively clearing debris. (1) Our finding that glial CED-10 levels can modulate engulfment rates, NRE shape, and animal behavior suggests that this process can be triggered by glia. (2) While both CED-10 overexpression and *ttx-1* mutants have short NRE (*Satterlee et al., 2001*; *Figure 4—figure supplement 1B*), unlike animals overexpressing CED-10, *ttx-1* mutants have fewer puncta, not more (*Figure 2A*). Thus, short NRE shape can derive from independent mechanisms. (3) While both *ced-10* and *tax-2* mutants have longer, disorganized NRE (*Satterlee et al., 2004*; *Singhvi et al., 2016*), *tax-2* mutants have more puncta, not fewer. If glial pruning only passively cleared debris, we would have expected the opposite.

Furthermore, that engulfment tracks neuron activity and modulating this process impacts animal behavior also suggests a physiological role for this process.

In summary, our findings reveal glial engulfment as an active regulator of neural functions. Importantly, they directly and causally link pruning of individual neuron endings to animal behavior at single-molecule and single-cell resolution. This raises the possibility that engulfment may be a general mechanism by which glia dynamically modulate sensory perception and neural functions, across modalities, systems, and species.

# Materials and methods

## Worm methods

*C. elegans* animals were cultured as previously described (*Brenner, 1974*; *Stiernagle, 2006*). Bristol N2 strain was used as wild type. For all experiments, animals were raised at 20°C for at least two generations without starvation, picked as L4 larvae onto fresh plate, and assayed 1 day later unless otherwise noted. Germline transformations by microinjection to generate unstable extrachromosomal array transgenes were carried out using standard protocols (*Fire et al., 1990*; *Mello et al., 1991*; *Stinchcomb et al., 1985*). Integration of extrachromosomal arrays was performed using UV+ trimethyl psoralen. All transgenic arrays were generated with 5 ng/μl $P_{elt-2:mCherry}$, 20 ng/μl $P_{mig-24}$:*Venus*, or 20 ng/μL $P_{unc-122}$:*RFP* as co-injection markers (*Abraham et al., 2007*; *Armenti et al., 2014*; *Miyabayashi et al., 1999*). Further information on all genetic strains and reagents is available upon request.

## Plasmids

### CED-10 plasmids

ced-10B isoform cDNA was isolated from a mixed stage cDNA library by PCR amplification with primers containing *Xma*I and *Nhe*I restriction enzyme sites and directionally ligated into pAS465 ($P_{F53F4.13}$:*SL2:mCherry*) to generate pAS275 plasmid. CED-10$^{G12V}$ and CED-10$^{T17N}$ mutations were derived by site-directed mutagenesis of pAS275 plasmid to produce pASJ29 (pSAR8) and pASJ37 (pSAR11), respectively.

### CED-12 plasmids

ced-12B isoform cDNA was isolated from a mixed stage cDNA library by PCR amplification with primers containing a *Xma*I and *Nhe*I restriction enzyme sites and directionally ligated into pAS465 to generate the pASJ11 (pSAR1) plasmid.

### PSR-1 plasmid

psr-1 C isoform cDNA was isolated from a mixed stage cDNA library by PCR amplification with primers containing *Bam*HI and *Nhe*I restriction enzyme sites and directionally ligated into pAS465 to generate the pASJ23 (pSAR7) plasmid.

### TAT-1 plasmid

tat-1 A isoform cDNA was generously gifted by the lab of Ding Xue. The $P_{SRTX-1b}$ promoter fragment was digested from the pSAR19 plasmid with *Sph*I and *Xma*I. A 430 bp fragment of the genomic *tat-1* sequence containing the first two exons and first intron was amplified by PCR with added 5′ *Xma*I site. This fragment was digested with *Xma*I and *Sph*I. The p49_78 plasmid containing *tat-1* cDNA was digested with *Sph*I, and all three fragments were ligated to make pASJ114 (pSAR35). Correct orientation was confirmed by sequencing of the ligation product.

### GFP:PSR-1 plasmid

psr-1 C isoform cDNA was isolated from a mixed stage cDNA library by PCR amplification with primers containing *Bam*HI and *Pst*I restriction enzyme sites and ligated into pAS516 ($P_{F53F4.13}$:*GFP*) to produce pASJ56 (pSAR18).

### His-Cl1 PLASMID

Histamine gated chloride channel sequence from pNP424 (*Pokala et al., 2014*) was restriction digested with *Nhe*I and *Kpn*I enzymes and ligated to pAS178 (P*SRTX-1*:SL2:GFP) to produce pAS540.

### Recombineered fosmids

The following fosmids with GFP recombineered in-frame in the coding sequence were obtained from the MPI-TransgeneOme Project: *gcy-8* (Clone ID: 02097061181003035 C08), *gcy-18* (Clone ID: 9735267524753001 E03), and *gcy-23* (Clone ID: 6523378417130642 E08).

## Microscopy, image processing, and analyses

Animals were immobilized using either 2 mM tetramizole or 100 nm polystyrene beads (Bangs Laboratories, catalog # PS02004). Images were collected on a Deltavision Elite RoHS wide-field deconvolution system with Ultimate Focus (GE), a PlanApo 60×/1.42 NA or OLY 100×/1.40 NA oil-immersion objective and a DV Elite CMOS Camera. Super-resolution microscopy images were collected on the Leica VT-iSIM microscope or the Leica SP8 confocal with Lightning. Images were processed on ImageJ, Adobe Photoshop CC, or Adobe Illustrator CC.

Binning categories for population analyses were based on preliminary analyses of population distribution of puncta numbers/animal in wild type, and mutants with excess puncta (*tax-2*) or reduced puncta mutants (*ced-10, psr-1*). Preliminary analyses of these strains suggested that the bin intervals (0, 1–9, or 10+ puncta) are the most robust, conservative, and rapid assessment of phenotypes. Higher than 10 puncta/cell were not readily resolved without post-processing and therefore binned together in population scores. Some genotypes were selected for further *post-hoc* single-cell puncta quantification analyses. For this, glia puncta numbers were quantified using Analyze Particles function in ImageJ on deconvolved images. Individual puncta size measurements were done on yz orthogonal rendering of optical sections using 3D objects counter plug-in in ImageJ.

## Electron microscopy

Adult hermaphrodites were fixed in 0.8% glutaraldehyde−0.8% osmium tetroxide−0.1 M cacodylate buffer (pH 7.4) for 1 hr at 4˚C in the dark, and then quickly rinsed several times with 0.1 M cacodylate buffer. Animal heads were decapitated and fixed in 1% osmium tetroxide−0.1 M cacodylate buffer overnight at 4˚C, quickly rinsed several times in 0.1 M cacodylate buffer, and dehydrated through a graded ethanol series. The samples were then embedded in Eponate 12 resin (Ted Pella, Inc, Redding, CA) and polymerized overnight in a 60˚C oven. 70 nm ultrathin serial sections were collected onto pioloform-coated slot grids from the anterior tip of the animal to a distance of approximately 7 μm. Sections were examined on a JEOL 1400 TEM (JEOL, Tokyo, Japan) at an accelerating voltage of 120 kV. Images were acquired with a Gatan Rio 4k × 4k detector (Gatan, Inc, Pleasanton, CA). Microvilli size measurements were done with ImageJ Measure Function on electron micrograph thin sections.

## Statistical analyses

Population puncta scoring was statistically analyzed using Fisher's exact statistical test in GraphPad Prism 8. Puncta images were quantified using Analyze Particles function in Image J and analyzed with a one-Way ANOVA with multiple-comparison test in GraphPad Prism 8.

## Chemogenetic silencing and RNAi

For chemogenetic silencing assays, 10 mM histamine (Sigma, catalog # H7250) was added to NGM agar plates. L4 larval stage transgenic worms expressing HisCl1 in AFD were grown for 24 hr on either normal or histamine plates and assayed as day 1 adults (*Pokala et al., 2014*). Plasmids expressing double-stranded RNA were obtained from the Ahringer Library (*Fraser et al., 2000*; *Kamath and Ahringer, 2003*). The L4440 empty vector was used as negative control. RNAi was performed by feeding synchronized L1 animals RNAi bacteria (*Timmons, 2004*). L4 larva were moved to a fresh plate with RNAi bacteria and scored 24 hr later for glial puncta (*nsIs483)* or AFD–NRE defects (*nsIs645).*

## Animal behavior assays

Thermotaxis assays were performed on a 17−26°C linear temperature gradient, designed as previously described (*Hedgecock and Russell, 1975*; *Mori and Ohshima, 1995*). Animals were synchronized and the staged progeny were tested on the first day of adulthood. Briefly, animals were washed twice with S-Basal and spotted onto the center of a 10 cm plate warmed to room temperature and containing 12 ml of NGM agar. The plate was placed onto the temperature gradient (17–26°C) with the addition of 5 ml glycerol to its bottom to improve thermal conductivity. At the end of 45 min, the plate was inverted over chloroform to kill the animals and allow easy counting of animals in each bin. The plates have an imprinted 6 × 6 square pattern, which formed the basis of the six temperature bins. Each data point is the average of 3–8 assays with ~150 worms/assay.

## Acknowledgements

We thank members of the Singhvi Laboratory, Harmit Malik, Jihong Bai, and Linda Buck for discussions and comments on the manuscript, and reviewers for their thoughtful comments. We apologize to those whose work was not cited due to our oversight or due to space considerations.

## Additional information

### Funding

| Funder | Grant reference number | Author |
|---|---|---|
| Simons Foundation | New Investigator Award | Aakanksha Singhvi |
| American Federation for Aging Research | New Investigator Award | Aakanksha Singhvi |
| NIH Office of the Director | R35NS105094 | Shai Shaham |
| NIH Office of the Director | NS114222 | Aakanksha Singhvi |
| NIH Office of the Director | T32AG066574 | Stephan Raiders |

The funders had no role in study design, data collection and interpretation, or the decision to submit the work for publication.

### Author contributions

Stephan Raiders, Conceptualization, Data curation, Formal analysis, Investigation, Methodology, Writing - original draft, Writing - review and editing; Erik Calvin Black, Data curation, Investigation, Methodology, Writing - review and editing; Andrea Bae, Stephen MacFarlane, Investigation, Methodology; Mason Klein, Resources; Shai Shaham, Writing - review and editing, Initial experimental designs; Aakanksha Singhvi, Conceptualization, Resources, Data curation, Formal analysis, Supervision, Funding acquisition, Validation, Investigation, Visualization, Methodology, Writing - original draft, Project administration, Writing - review and editing

### Author ORCIDs

Stephan Raiders https://orcid.org/0000-0002-2394-9193
Mason Klein https://orcid.org/0000-0001-8211-077X
Shai Shaham http://orcid.org/0000-0002-3751-975X
Aakanksha Singhvi https://orcid.org/0000-0001-5782-8536

### Decision letter and Author response

Decision letter https://doi.org/10.7554/eLife.63532.sa1
Author response https://doi.org/10.7554/eLife.63532.sa2

## Additional files

### Supplementary files
• Transparent reporting form

### Data availability
All data generated in this study are included in the manuscript and supporting files. Reagents are available from the corresponding author upon reasonable request.

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

# Appendix 1

**Appendix 1—key resources table**

| Reagent type (species) or resource | Designation | Source or reference | Identifiers | Additional information |
|---|---|---|---|---|
| Strain, strain background (*Caenorhabditis elegans*) | *nsIs481* | This paper | Singhvi Lab Database ID: OS8556 | [20 ng/µl 02097061181003035 C08 ($P_{gcy-8}$:gcy-8:GFP) + $P_{elt-2}$:mCherry]. Integration of nsEx3945. Request from corresponding author. |
| Strain, strain background (*C. elegans*) | *nsIs482* | This paper | Singhvi Lab Database ID: OS8557 | [20 ng/µl 02097061181003035 C08 ($P_{gcy-8}$:gcy-8:GFP) + $P_{elt-2}$:mCherry]. Integration of nsEx3945. Request from corresponding author. |
| Strain, strain background (*C. elegans*) | *nsIs483 X* | This paper | Singhvi Lab Database ID: OS8558 | [20 ng/µl 02097061181003035 C08 ($P_{gcy-8}$:gcy-8:GFP) + $P_{elt-2}$:mCherry]. Integration of nsEx3945. Request from corresponding author. |
| Strain, strain background (*C. elegans*) | *nsIs484* | This paper | Singhvi Lab Database ID: OS8502 | [20 ng/µl 02097061181003035 C08 ($P_{gcy-8}$:gcy-8:GFP) + $P_{elt-2}$:mCherry]. Integration of nsEx3945. Request from corresponding author. |
| Strain, strain background (*C. elegans*) | *nsIs645 IV* | This paper | Singhvi Lab Database ID: OS10884 | [50 ng/µl pAS322 ($P_{srtx-1B}$:STRX-1:GFP) + $P_{unc-122}$:RFP]. Integration of nsEx4078. Request from corresponding author. |
| Strain, strain background (*C. elegans*) | *nsIs647* | This paper | Singhvi Lab Database ID: OS10805 | [50 ng/µl pAS322 ($P_{srtx-1B}$:STRX-1:GFP) + $P_{unc-122}$:RFP]. Integration of nsEx4078. Request from corresponding author. |
| Strain, strain background(*C. elegans*) | *dnaIs1* | This paper | Singhvi Lab Database ID: ASJ160 | [50 ng/µl pAS540 ($P_{srtx-1B}$:HisCl1:SL2:GFP) + $_{elt-2}$:mCherry]. Integration of nsEx5340. Request from corresponding author. |
| Strain, strain background(*C. elegans*) | *dnaIs2* | This paper | Singhvi Lab Database ID: ASJ161 | [50 ng/µl pAS540 ($P_{srtx-1B}$:HisCl1:SL2:GFP) + $_{elt-2}$:mCherry]. Integration of nsEx5340. Request from corresponding author. |
| Strain, strain background (*C. elegans*) | *dnaIs3* | This paper | Singhvi Lab Database ID: ASJ271 | [50 ng/µl pAS540 ($P_{srtx-1B}$:HisCl1:SL2:GFP) + $_{elt-2}$:mCherry]. Integration of nsEx5340. Request from corresponding author. |
| Strain, strain background (*C. elegans*) | *dnaIs4* | This paper | Singhvi Lab Database ID: ASJ280 | [50 ng/µl pAS540 ($P_{srtx-1B}$:HisCl1:SL2:GFP) + $_{elt-2}$:mCherry]. Integration of nsEx5340. Request from corresponding author. |
| Strain, strain background (*C. elegans*) | *dnaIs7* | This paper | Singhvi Lab Database ID: ASJ360 | [5 ng/µl pAS275 ($P_{F53F4.13}$:CED-10:SL2:mCherry) + $P_{mig-24}$:Venus]. Integration of nsEx5365. Request from corresponding author. |
| Strain, strain background (*C. elegans*) | *dnaIs8* | This paper | Singhvi Lab Database ID: ASJ359 | [5 ng/µl pAS275 ($P_{F53F4.13}$:CED-10:SL2:mCherry) + $P_{mig-24}$:Venus]. Integration of nsEx5365. Request from corresponding author. |
| Strain, strain background (*C. elegans*) | *nsIs143X* | *Procko et al., 2011* | OS9176 | $P_{F16F9.3}$:DsRed. |
| Strain, strain background (*C. elegans*) | *nsIs109* | *Bacaj et al., 2008b* | OS1932 | $P_{F16F9.3}$:DTA(G53E). |
| Strain, strain background (*C. elegans*) | *nsEx3944* | *Singhvi et al., 2016* | Singhvi Lab Database ID: OS7171 | [20 ng/µl 02097061181003035 C08 ($P_{gcy-8}$:gcy-8:GFP) + $P_{elt-2}$:mCherry]. Request from either corresponding author or Dr. Shai Shaham (The Rockefeller University, USA). |

*Continued on next page*

*Appendix 1—key resources table continued*

| Reagent type (species) or resource | Designation | Source or reference | Identifiers | Additional information |
|---|---|---|---|---|
| Strain, strain background (*C. elegans*) | nsEx3945 | *Singhvi et al., 2016* | Singhvi Lab Database ID: OS7172 | [20 ng/µl 02097061181003035 C08 ($P_{gcy-8}$: gcy-8:GFP) + $P_{elt-2}$:mCherry]. Request from either corresponding author or Dr. Shai Shaham (The Rockefeller University, USA). |
| Strain, strain background (*C. elegans*) | nsEx3946 | *Singhvi et al., 2016* | Singhvi Lab Database ID: OS7173 | [20 ng/µl 02097061181003035 C08 ($P_{gcy-8}$: gcy-8:GFP) + $P_{elt-2}$:mCherry]. Request from either corresponding author or Dr. Shai Shaham (The Rockefeller University, USA). |
| Strain, strain background (*C. elegans*) | nsEx3947 | *Singhvi et al., 2016* | Singhvi Lab Database ID: OS7174 | [20 ng/µl 02097061181003035 C08 ($P_{gcy-8}$: gcy-8:GFP) + $P_{elt-2}$:mCherry]. Request from either corresponding author or Dr. Shai Shaham (The Rockefeller University, USA). |
| Strain, strain background (*C. elegans*) | nsEx4733 | This paper | Singhvi Lab Database ID: OS9078 | [20 ng/µl 9735267524753001 E03 ($P_{gcy-18}$: gcy-18:GFP) + $P_{elt-2}$:mCherry]. Request from corresponding author. |
| Strain, strain background (*C. elegans*) | nsEx4734 | This paper | Singhvi Lab Database ID: OS9079 | [20 ng/µl 9735267524753001 E03 ($P_{gcy-18}$: gcy-18:GFP) + $P_{elt-2}$:mCherry]. Request from corresponding author. |
| Strain, strain background (*C. elegans*) | nsEx4857 | This paper | Singhvi Lab Database ID: OS9406 | [20 ng/µl 9735267524753001 E03 ($P_{gcy-18}$: gcy-18:GFP) + $P_{elt-2}$:mCherry]. Request from corresponding author. |
| Strain, strain background (*C. elegans*) | nsEx4763 | This paper | Singhvi Lab Database ID: OS9164 | [20 ng/µl 9735267524753001 E03 ($P_{gcy-18}$: gcy-18:GFP) + $P_{elt-2}$:mCherry]. Request from corresponding author. |
| Strain, strain background (*C. elegans*) | nsEx4803 | This paper | Singhvi Lab Database ID: OS9276 | [20 ng/µl 6523378417130642 E08 ($P_{gcy-23}$: gcy-23:GFP) + $P_{elt-2}$:mCherry]. Request from corresponding author. |
| Strain, strain background (*C. elegans*) | nsEx4765 | This paper | Singhvi Lab Database ID: OS9166 | [20 ng/µl 6523378417130642 E08 ($P_{gcy-23}$: gcy-23:GFP) + $P_{elt-2}$:mCherry]. Request from corresponding author. |
| Strain, strain background (*C. elegans*) | nsEx4392 | This paper | Singhvi Lab Database ID: OS8257 | [20 ng/µl pAS428 ($P_{srtx-1B}$:DYF-11:GFP) + $P_{elt-2}$:mCherry]. Request from corresponding author. |
| Strain, strain background (*C. elegans*) | nsEx4393 | This paper | Singhvi Lab Database ID: OS8258 | [20 ng/µl pAS428 ($P_{srtx-1B}$:DYF-11:GFP) + $P_{elt-2}$:mCherry]. Request from corresponding author. |
| Strain, strain background (*C. elegans*) | nsEx4394 | This paper | Singhvi Lab Database ID: OS8259 | [20 ng/µl pAS428 ($P_{srtx-1B}$:DYF-11:GFP) + $P_{elt-2}$:mCherry]. Request from corresponding author. |
| Strain, strain background (*C. elegans*) | nsEx4446 | This paper | Singhvi Lab Database ID: OS8330 | [20 ng/µl pAS428 ($P_{srtx-1B}$:DYF-11:GFP) + $P_{elt-2}$:mCherry]. Request from corresponding author. |
| Strain, strain background (*C. elegans*) | nsEx4051 | This paper | Singhvi Lab Database ID: OS7443 | [50 ng/µl pAS322 ($P_{srtx-1B}$:SRTX-1:GFP) + $P_{unc-122}$:RFP]. Request from corresponding author. |
| Strain, strain background (*C. elegans*) | nsEx4077 | This paper | Singhvi Lab Database ID: OS7541 | [50 ng/µl pAS322 ($P_{srtx-1B}$:SRTX-1:GFP) + $P_{unc-122}$:RFP]. Request from corresponding author. |
| Strain, strain background (*C. elegans*) | nsEx4078 | This paper | Singhvi Lab Database ID: OS7542 | [50 ng/µl pAS322 ($P_{srtx-1B}$:SRTX-1:GFP) + $P_{unc-122}$:RFP]. Request from corresponding author. |
| Strain, strain background (*C. elegans*) | nsEx4570 | This paper | Singhvi Lab Database ID: OS8598 | [25 ng/µl pAS447 ($P_{srtx-1}$:EGL-1) + $P_{mig-24}$: Venus]. Request from corresponding author. |

*Continued on next page*

*Appendix 1—key resources table continued*

| Reagent type (species) or resource | Designation | Source or reference | Identifiers | Additional information |
|---|---|---|---|---|
| Strain, strain background (*C. elegans*) | *nsEx4616* | This paper | Singhvi Lab Database ID: OS8767 | [25 ng/µl pAS447 ($P_{srtx-1}$:*EGL-1*) + $P_{mig-24}$:Venus]. Request from corresponding author. |
| Strain, strain background (*C. elegans*) | *nsEx4688* | This paper | Singhvi Lab Database ID: OS8970 | [25 ng/µl pAS447 ($P_{srtx-1}$:*EGL-1*) + $P_{mig-24}$:Venus]. Request from corresponding author. |
| Strain, strain background (*C. elegans*) | *nsEx5266* | This paper | Singhvi Lab Database ID: OS10640 | [50 ng/µl pAS540 ($P_{srtx-1}$:*HisCl1:SL2:GFP*) + $P_{elt-2}$:*mCherry*]. Request from corresponding author. |
| Strain, strain background (*C. elegans*) | *nsEx5340* | This paper | Singhvi Lab Database ID: OS10735 | [50 ng/µl pAS540 ($P_{srtx-1}$:*HisCl1:SL2:GFP*) + $P_{elt-2}$:*mCherry*]. Request from corresponding author. |
| Strain, strain background (*C. elegans*) | *nsEx5356* | This paper | Singhvi Lab Database ID: OS10761 | [50 ng/µl pAS540 ($P_{srtx-1}$:*HisCl1:SL2:GFP*) + $P_{elt-2}$:*mCherry*]. Request from corresponding author. |
| Strain, strain background (*C. elegans*) | *nsEx5365* | This paper | Singhvi Lab Database ID: OS10781 | [5 ng/µl pAS275 ($P_{F53F4.13}$:*CED-10B:SL2: mCherry*) + $P_{mig-24}$:Venus]. Request from corresponding author. |
| Strain, strain background (*C. elegans*) | *nsEx5381* | This paper | Singhvi Lab Database ID: OS10826 | [5 ng/µl pAS275 ($P_{F53F4.13}$:*CED-10B:SL2: mCherry*) + $P_{mig-24}$:Venus]. Request from corresponding author. |
| Strain, strain background (*C. elegans*) | *nsEx5382* | This paper | Singhvi Lab Database ID: OS10877 | [5 ng/µl pAS275 ($P_{F53F4.13}$:*CED-10B:SL2: mCherry*) + $P_{mig-24}$:Venus]. Request from corresponding author. |
| Strain, strain background (*C. elegans*) | *dnaEx1* | This paper | Singhvi Lab Database ID: ASJ06 | [5 ng/µl pASJ11-pSAR1 ($P_{F53F4.13}$:*CED-12B: SL2:mCherry*) + $P_{unc-122}$:RFP]. Request from corresponding author. |
| Strain, strain background (*C. elegans*) | *dnaEx2* | This paper | Singhvi Lab Database ID: ASJ07 | [5 ng/µl pASJ11-pSAR1 ($P_{F53F4.13}$:*CED-12B: SL2:mCherry*) + $P_{unc-122}$:RFP]. Request from corresponding author. |
| Strain, strain background (*C. elegans*) | *dnaEx3* | This paper | Singhvi Lab Database ID: ASJ08 | [5 ng/µl pASJ11-pSAR1 ($P_{F53F4.13}$:*CED-12B: SL2:mCherry*) + $P_{unc-122}$:RFP]. Request from corresponding author. |
| Strain, strain background (*C. elegans*) | *dnaEx19* | This paper | Singhvi Lab Database ID: ASJ104 | [5 ng/µl pASJ23-pSAR7 ($P_{F53F4.13}$:*PSR-1C: SL2:mCherry*) + $P_{mig-24}$:Venus]. Request from corresponding author. |
| Strain, strain background (*C. elegans*) | *dnaEx30* | This paper | Singhvi Lab Database ID: ASJ143 | [5 ng/µl pASJ23-pSAR7 ($P_{F53F4.13}$:*PSR-1C: SL2:mCherry*) + $P_{mig-24}$:Venus]. Request from corresponding author. |
| Strain, strain background (*C. elegans*) | *dnaEx33* | This paper | Singhvi Lab Database ID: ASJ147 | [5 ng/µl pASJ23-pSAR7 ($P_{F53F4.13}$:*PSR-1C: SL2:mCherry*) + $P_{mig-24}$:Venus]. Request from corresponding author. |
| Strain, strain background (*C. elegans*) | *dnaEx29* | This paper | Singhvi Lab Database ID: ASJ142 | [5 ng/µl pASJ29-pSAR8 ($P_{F53F4.13}$:*CED-$10B^{G12V}$:SL2:mCherry*) + $P_{unc-122}$:RFP]. Request from corresponding author. |
| Strain, strain background (*C. elegans*) | *dnaEx51* | This paper | Singhvi Lab Database ID: ASJ218 | [5 ng/µl pASJ37 (pSAR11) ($P_{F53F4.13}$:*CED-$10B^{T17N}$:SL2:mCherry*) + $P_{unc-122}$:RFP]. Request from corresponding author. |
| Strain, strain background (*C. elegans*) | *dnaEx57* | This paper | Singhvi Lab Database ID: ASJ225 | [5 ng/µl pASJ37 (pSAR11) ($P_{F53F4.13}$:*CED-$10B^{T17N}$:SL2:mCherry*) + $P_{unc-122}$:RFP]. Request from corresponding author. |
| Strain, strain background (*C. elegans*) | *dnaEx59* | This paper | Singhvi Lab Database ID: ASJ230 | [5 ng/µl pASJ37 (pSAR11) ($P_{F53F4.13}$:*CED-$10B^{T17N}$:SL2:mCherry*) + $P_{unc-122}$:RFP]. Request from corresponding author. |

*Appendix 1—key resources table continued*

| Reagent type (species) or resource | Designation | Source or reference | Identifiers | Additional information |
|---|---|---|---|---|
| Strain, strain background (*C. elegans*) | *nsEx5268* | This paper | Singhvi Lab Database ID: OS10642 | [5 ng/µl pAS247 ($P_{F53F4.13}$: *WSP-1:SL2:mCherry*) + $P_{mig-24}$:Venus]. Request from corresponding author. |
| Strain, strain background (*C. elegans*) | *nsEx5363* | This paper | Singhvi Lab Database ID: OS10779 | [5 ng/µl pAS247 ($P_{F53F4.13}$:*WSP-1:SL2:mCherry*) + $P_{mig-24}$:Venus]. Request from corresponding author. |
| Strain, strain background (*C. elegans*) | *nsEx5380* | This paper | Singhvi Lab Database ID: OS10825 | [5 ng/µl pAS247 ($P_{F53F4.13}$:*WSP-1:SL2:mCherry*) + $P_{mig-24}$:Venus]. Request from corresponding author. |
| Strain, strain background (*C. elegans*) | *dnaEx160* | This paper | Singhvi Lab Database ID: ASJ488 | [45 ng/µl pASJ114-pSAR35 ($P_{srtx-1B}$:*TAT-1A*) + $P_{unc-122}$:RFP]. Request from corresponding author. |
| Strain, strain background (*C. elegans*) | *dnaEx162* | This paper | Singhvi Lab Database ID: ASJ498 | [45 ng/µl pASJ114-pSAR35 ($P_{srtx-1B}$:*TAT-1A*) + $P_{unc-122}$:RFP]. Request from corresponding author. |
| Strain, strain background (*C. elegans*) | *dnaEx70* | This paper | Singhvi Lab Database ID: ASJ266 | [2.5 ng/µl pASJ56-pSAR18 ($P_{F53F4.13}$:*GFP:PSR-1C*) + $P_{unc-122}$:RFP]. Request from corresponding author. |
| Strain, strain background (*C. elegans*) | *dnaEx71* | This paper | Singhvi Lab Database ID: ASJ267 | [2.5 ng/µl pASJ56-pSAR18 ($P_{F53F4.13}$:*GFP:PSR-1C*) + $P_{unc-122}$:RFP]. Request from corresponding author. |
| Strain, strain background (*C. elegans*) | *dnaEx74* | This paper | Singhvi Lab Database ID: ASJ273 | [2.5 ng/µl pASJ56-pSAR18 ($P_{F53F4.13}$:*GFP:PSR-1C*) + $P_{unc-122}$:RFP]. Request from corresponding author. |
| Strain, strain background (*C. elegans*) | Wild type | CGC | Singhvi Lab Database ID: N2 | Reference strain. |
| Strain, strain background (*C. elegans*) | *tax-2(p691)* I | CGC | Singhvi Lab Database ID: PR691 | |
| Strain, strain background (*C. elegans*) | *ced-12 (n3261)* I | CGC | Singhvi Lab Database ID: MT11068 | |
| Strain, strain background (*C. elegans*) | *ced-12(k149)* I | CGC | Singhvi Lab Database ID: NF87 | |
| Strain, strain background (*C. elegans*) | *psr-1(tm469)* I | CGC | Singhvi Lab Database ID: CU1715 | |
| Strain, strain background (*C. elegans*) | *ced-1 (e1754)* I | CGC | Singhvi Lab Database ID: CB3261 | |
| Strain, strain background (*C. elegans*) | *ced-1 (e1735)* I | CGC | Singhvi Lab Database ID: CB3203 | |
| Strain, strain background (*C. elegans*) | *unc-73(e936)* I | CGC | Singhvi Lab Database ID: CB936 | |
| Strain, strain background (*C. elegans*) | *scrm-1 (tm805)* I | CGC | Singhvi Lab Database ID: CU2945 | |
| Strain, strain background (*C. elegans*) | *ttr-52 (tm2078)* III | NBRP | Singhvi Lab Database ID: FX002078 | *Kang et al., 2012*. |

*Continued on next page*

*Appendix 1—key resources table continued*

| Reagent type (species) or resource | Designation | Source or reference | Identifiers | Additional information |
|---|---|---|---|---|
| Strain, strain background (*C. elegans*) | *ced-6 (n1813)* III | CGC | Singhvi Lab Database ID: MT4433 | |
| Strain, strain background (*C. elegans*) | *tat-1 (tm1034)* III | NBRP | Singhvi Lab Database ID: FX001034 | *Darland-Ransom et al., 2008*. |
| Strain, strain background (*C. elegans*) | *tax-4(p678)* III | CGC | Singhvi Lab Database ID: PR678 | |
| Strain, strain background (*C. elegans*) | *ced-7 (n2094)* III | CGC | Singhvi Lab Database ID: MT8886 | |
| Strain, strain background (*C. elegans*) | *ver-1 (ok1738)* III | CGC | Singhvi Lab Database ID: VC1263 | *Consortium, C.e.D.M, 2012*. |
| Strain, strain background (*C. elegans*) | *ver-2(ok897)* III | CGC | Singhvi Lab Database ID: RB983 | *Consortium, C.e.D.M, 2012*. |
| Strain, strain background (*C. elegans*) | *ina-1 (gm144)* III | CGC | Singhvi Lab Database ID: NG144 | |
| Strain, strain background (*C. elegans*) | *ced-10 (n3246)*IV | CGC | Singhvi Lab Database ID: MT9958 | |
| Strain, strain background (*C. elegans*) | *ced-10 (n1993)* IV | CGC | Singhvi Lab Database ID: MT5013 | |
| Strain, strain background (*C. elegans*) | *ced-2 (e1752)* IV | CGC | Singhvi Lab Database ID: CB3257 | |
| Strain, strain background (*C. elegans*) | *ced-5 (n1812)* IV | CGC | Singhvi Lab Database ID: MT4434 | |
| Strain, strain background (*C. elegans*) | *cng-3(jh113)* IV | CGC | Singhvi Lab Database ID: KJ462 | |
| Strain, strain background (*C. elegans*) | *ttx-1(p767)* V | CGC | Singhvi Lab Database ID: PR767 | |
| Strain, strain background (*C. elegans*) | *osm-6(p811)* V | CGC | Singhvi Lab Database ID: PR811 | |
| Strain, strain background (*C. elegans*) | *dyf-11 (mn392)* X | CGC | Singhvi Lab Database ID: SP1713 | |
| Strain, strain background (*C. elegans*) | *ced-8 (n1891)* X | CGC | Singhvi Lab Database ID: MT5006 | |
| Strain, strain background (*C. elegans*) | *ver-3(ok891)* X | CGC | Singhvi Lab Database ID: VC610 | *Consortium, C.e.D.M, 2012*. |
| Strain, strain background (*C. elegans*) | *ver-4 (ok1079)* X | CGC | Singhvi Lab Database ID: RB1100 | *Consortium, C.e.D.M, 2012*. |

*Appendix 1—key resources table continued*

| Reagent type (species) or resource | Designation | Source or reference | Identifiers | Additional information |
|---|---|---|---|---|
| Strain, strain background (*C. elegans*) | *egl-15(n484) X* | CGC | Singhvi Lab Database ID: OS10586 | |
| Genetic reagent (*Escherichia coli*) | *pat-2 RNAi* | ***Kamath and Ahringer, 2003*** | Singhvi Lab Database ID: pASJ_RNAi_1D1 | Ahringer RNAi library: WBGene00018832. |

