## [Decision Letter]

**Acceptance summary:**

In mammals, glial cells (abundant, non-neuronal components of the nervous system) can engulf and sculpt the receptive endings of neurons, a function that is important for maintenance and plasticity of nervous system function. Here, Raiders et al. find that a glial cell in the nematode *C. elegans* can actively sculpt nerve receptive endings; further, they identify key components of the mechanism by which this takes place and show that it is important for proper neuronal function and behavior. Thus, their studies indicate that pruning of nerve receptive endings may be an ancient and conserved function of glial cells.

**Decision letter after peer review:**

Thank you for submitting your article "Glia actively sculpt sensory neurons by controlled phagocytosis to tune animal behavior" for consideration by *eLife*. Your article has been reviewed by 3 peer reviewers, including Douglas Portman as the Reviewing Editor and Reviewing #1, and the evaluation has been overseen by Piali Sengupta as the Senior Editor. The following individual involved in review of your submission has agreed to reveal their identity: Maureen M Barr (Reviewer #2).

The reviewers have discussed the reviews with one another and the Reviewing Editor has drafted this decision to help you prepare a revised submission.

Summary:

This interesting manuscript reports the observation that the *C. elegans* AMsh glial cell can engulf bits of villi from the elaborate sensory endings of the AFD thermosensory neurons and that it may be able to exert active regulation of this process. Using a powerful combination of genetic analysis, imaging, and behavior, the study proposes a mechanism by which exposed PS phospholipids on the AFD villi serve as a signal for a psr-1/ced-10/wsp-1-dependent engulfment pathway. The paper provides evidence consistent with the possibility (but does not definitively show) that the extent of engulfment may be related to AFD activity, and that compromising the engulfment pathway changes the morphology of the AFD villi as well as thermosensory behavior. The parallels of this process with the engulfment functions of glia in the vertebrate CNS are intriguing and will attract the attention of glial biologists as well as sensory neurobiologists.

Overall, the reviewers found the paper to be exciting and potentially appropriate for *eLife*. However, the paper requires additional experiments to test particular aspects of the model as well as some significant re-writing and re-organization.

Essential revisions:

1. Several aspects of your model need to be more explicitly demonstrated. In particular, you will need to:

a. Determine whether psr-1 is expressed in AMsh, using a reporter containing its native promoter. Ideally this would be done using a CRISPR-based approach.

b. Determine whether tat-1 functions cell-autonomously in AFD.

c. Determine whether tax-4 (or cng-3 or tax-2) functions cell-autonomously in AFD.

d. Construct and analyze psr-1; tat-1 double mutants.

2. In a number of cases, the conclusions drawn from the experiments presented are not well justified. These need to be addressed either by carrying out additional experiments or tempering the claims made in the manuscript.

a. Based on several experiments, your paper concludes that AMsh determines the extent of engulfment of AFD villi fragments based on the level of activity of AFD. However, alternative explanations are not ruled out, particularly that activity alters the morphology of the AFD microvilli, which then indirectly alters engulfment. Please rewrite the manuscript to temper your claims or provide new evidence to support them.

b. On lines 366-7 and elsewhere in the paper, you conclude that pruning of villi by AMsh regulates thermosensation. This connection is tenuous; the changes in behavior you observe could result from other/indirect effects of the manipulations you carry out. Please tone down your interpretations or carry out additional experiments to substantiate them.

c. Several other places where conclusions need to be tempered or changed are listed below.

3. More details on statistical analysis are required. It's not always clear which tests were used for which sets of data, and there is no justification for using Kruskal-Wallis vs. Chi-squared tests. It's not clear whether corrections for multiple comparisons were carried out. In many of the figures, asterisks are placed in a way that makes it unclear which comparison they refer to.

4. The manuscript is rather difficult to read. The text repeatedly jumps back and forth between figures and panels, sometimes in a seemingly arbitrary way. Further, the text doesn't describe in sufficient detail exactly what is being shown in with regard to the many genetic manipulations carried out (alleles for example). The paper needs to be reorganized and rewritten to tighten the presentation of the data and make the story more accessible. It may also be useful to include a table summarizing all the candidate genes/pathways (including *C. elegans* and mammalian ortholog) and their phenotypes. Cartoon schematics to accompany the images would be useful for the non-*C. elegans* reader.

[Editors' note: further revisions were suggested prior to acceptance, as described below.]

Thank you for resubmitting your work entitled "Glia actively sculpt sensory neurons by controlled phagocytosis to tune animal behavior" for further consideration by *eLife*. Your revised article has been reviewed by 2 peer reviewers, including Doug Portman as the Reviewing Editor and the evaluation has been overseen by Piali Sengupta as the Senior Editor. The following individual involved in review of your submission has agreed to reveal their identity: Maureen Barr (Reviewer #2).

We find your resubmitted manuscript significantly improved, with nearly all of the reviewers' concerns addressed. The manuscript is reorganized, easier to follow, and the model is largely supported by the data. However, there are some remaining issues that need to be addressed.

Many of these issues have to do with data interpretation – there are a number of cases in which alternative explanations are not ruled out, or where the data do not strongly support the conclusions drawn:

1) Lines 133-135: The size ranges discussed here are not quite "in the range" of each other, as the text indicates. Please soften this conclusion.

2) Line 145-6: A trivial possibility that's not ruled out is that more puncta are seen in older animals because the AFD NRE isn't fully developed until the adult. Again, please soften your conclusion here or provide supporting data.

3) Lines 160 on: It's possible that no cilium fragments are seen in glia simply because the cilium is far smaller than the microvilli, which are larger and many. Please temper your conclusion.

4) Line 176: more than just the NRE is lost in ttx-1 mutants. One possibility could be that AFD no longer has the machinery to signal engulfment, possibly by putting PS on its outer leaflet.

5) Lines 252-268: There are significant concerns about the interpretation of the results presented here, particularly the pat-2(RNAi) experiments. The text states that pat-2(RNAi) significantly blocks phagocytosis, but Figure 2G clearly shows that this is not the case. Further, many interaction tests are carried out using pat-2(RNAi) – any epistasis experiments using RNAi or non-null alleles need to be interpreted *extremely* cautiously, here and in the rest of the paper. This also leads to over-interpretation of the results of ced-1 experiments. Further, ced-1 hasn't been shown to function in AMsh, raising the possibility that its effects on puncta number could be indirect. As a result of these issues, the interpretations from this section don't seem particularly convincing or informative. This entire section could be removed without detracting from the advances made in your paper.

6) Lines 288-296: These experiments are difficult to interpret. The ced-10 phenotype is quite strong on its own, so floor effects get in the way of sorting out these genetic interactions. By themselves, these experiments don't convincingly show that ced-10 is downstream of ced-2/5/12 (though it may be legitimate to propose this based on previous findings). Similarly, the suppression of the psr-1 phenotype by ced-10 overexpression doesn't prove that ced-10 is downstream (it could act in parallel). Please soften your interpretation of these findings.

7) Please consider reinterpreting the relationship between neural activity and engulfment. AFD temperature responses have not shown to be defective in osm-6 mutants (or provide reference), so the changes you see here may be unrelated to activity. Also, at 25C, AFD is believed to have higher activity, but your results show that there are more puncta at 25C than at 15C.

---

## [Author Response]

Essential revisions:1. Several aspects of your model need to be more explicitly demonstrated. In particular, you will need to:a. Determine whether psr-1 is expressed in AMsh, using a reporter containing its native promoter. Ideally this would be done using a CRISPR-based approach.

We attempted to determine if *psr-1* is expressed in glia by generating six independent transgenic multi-copy array animal strains, driven by various *psr-1* regulatory sequence regions, as follows:

a. *Recombineered Fosmid* (3 different concentrations): This contained the entire *psr-1* genomic region fused in frame to a C-terminal GFP tag. At 50 ng/ul fosmid concentration, we could not recover Array+ animals after multiple attempts, hinting at possible toxicity from over-expression. At 35 ng/ul, some sickly transgenic animals were recovered, but no GFP was detectable at any stage. At 20 ng/ul, while the array+ animals recovered were healthier, there was no detectable GFP expression.

b. *P_psr-1B_:GFP transcriptional reporter (*2 different concentrations): A large 3^rd^ intron in the *psr-1* cds led us to wonder if it harbored regulatory sequences for PSR-1 expression (Author response image 1). We therefore generated a transcriptional reporter that includes the canonical sequence of the PSR-1A cDNA + ~1kb downstream sequence. We saw no detectable GFP expression at either 50ng/ul or 100ng/ul. Curiously, the difference between this and P_psr-1A_:GFP expression at equivalent concertation (see below), hints that the 3^rd^ intron may harbor potential repressor elements.

c. *P_psr-1A_:GFP transcriptional reporter:* At 100ng/ul, some embryo and larva express GFP in the head region of the animal, where AMsh glia and AFD are located (Author response image 1). This becomes rapidly restricted after L2 larva to a presumptive interneuron pair (Author response image 1, yellow arrow) and some variable neural cells of unclear lineage (Author response image 1 and E, magenta arrow). Thus, if PSR-1 expresses in AMsh glia (or AFD) after L4 larva, it remains undetectable by this approach. This is a confound because glial engulfment is not seen in animals younger than L4 larva (Figure 2C-D). Furthermore, we are not aware of any AMsh glia-specific embryonic promoter that does not also alter AFD NRE shape when expressed in multicopy arrays. Thus, even the tangential question of whether embryonic PSR-1 localizes to AMsh glia cannot be addressed by straightforward approaches.

We opted against CRISPR-based single-locus insertion because of concerns on low expression based on previous studies (Yang et al., *Nat Comm*, 2015; Abay et al., *PNAS*, 2017) and our results (Author response image 1). Other groups also failed to reveal PSR-1 expression with either antibody tagging or promoter expression. Indeed, to date, the endogenous PSR-1 expression pattern in *C. elegans* remains a mystery.

**Author response image 1. sa2fig1:** (A) *psr-1* genomic organization and transcriptional reporters tested. Gray: gene upstream; black = *psr-1* exons; green = GFP. (B-E). P_psr-1A_:GFP transcriptional reporter across different life stages, as noted on each panel. Presumptive neural cells in the head region of the animal (magenta arrow), and interneuron (yellow arrow) are seen. Scale bar = 5µm.

b. Determine whether tat-1 functions cell-autonomously in AFD.

We agree that this was an important missing data and thank the reviewer for this recommendation. We have now performed this experiment, and the data is presented in Figure 4B. Briefly, we expressed TAT-1 cDNA in *tat-1* mutants under the AFD-neuron specific promoter P_srtx-1B_ (Singhvi et al. 2016). This completely rescues the *tat-1* mutant defect. Thus, TAT-1 functions cell autonomously in the AFD neuron.

c. Determine whether tax-4 (or cng-3 or tax-2) functions cell-autonomously in AFD.

If we understand correctly, the concern is that other TAX-2(+) neurons may impact AMsh glial engulfment of AFD-NRE non-autonomously. While our *tax-2* results do not rule this out directly, we think that our finding that cell-specific chemo-genetic silencing of AFD neurons can also increase engulfment in a *ced-10* dependent manner, clearly indicate a role for AFD activity. This is also consistent with our TAT-1 rescue in AFD (Figure 4B). We also think that while it will be intriguing if other neurons non-autonomously drive engulfment, this will only add to our finding a role for AFD, but not counter this data. We have rephrased this section significantly, and hope this satisfies the concern.

d. Construct and analyze psr-1; tat-1 double mutants.

This experiment was a great suggestion. We constructed and analyzed *psr-1; tat-1* double mutants, and the data is now included in Figure 4D. Briefly, we find that the double mutant phenocopies *psr-1* single mutants. This suppression of *tat-1* increased puncta defects are consistent with our working model that *tat-1* functions upstream of *psr-1*.

2. In a number of cases, the conclusions drawn from the experiments presented are not well justified. These need to be addressed either by carrying out additional experiments or tempering the claims made in the manuscript.a. Based on several experiments, your paper concludes that AMsh determines the extent of engulfment of AFD villi fragments based on the level of activity of AFD. However, alternative explanations are not ruled out, particularly that activity alters the morphology of the AFD microvilli, which then indirectly alters engulfment. Please rewrite the manuscript to temper your claims or provide new evidence to support them.

This was our mis-communication. Briefly, we do not mean to suggest that engulfment is the SOLE mechanism to regulate AFD-NRE shape. Our data suggests that it is ONE mechanism. We, in fact, suspect that AMsh glia and AFD regulate NRE shape through additional glial and neuron pathways. In fact, we have previously uncovered an independent glia-neuron interaction mechanism (Singhvi et al., 2016). Our thesis in this manuscript is that glial engulfment tracks AFD activity and regulates of AFD NRE shape and animal thermosensory behavior. We have now rephrased this in both the Results and Discussion sections for clarity.

In this manuscript, we show that (a) AMsh glia engulf AFD NRE microvilli (Figure 1-3) (b) manipulating glial CED-10 alters engulfment (Figure 5), (c) this tracks AFD activity (Figure 6) and (d) regulates AFD NRE shape and animal behavior (Figure 7). Indeed, these and our other unpublished data lead us to suspect that additional mechanisms connect AFD activity cue to glial CED-10 regulation independent of PSR-1.

b. On lines 366-7 and elsewhere in the paper, you conclude that pruning of villi by AMsh regulates thermosensation. This connection is tenuous; the changes in behavior you observe could result from other/indirect effects of the manipulations you carry out. Please tone down your interpretations or carry out additional experiments to substantiate them.

Thank you for this astute comment. We agree that this was a caveat of our interpretation. Indeed, this would confound analyzing behavior defects in all molecules we had identified in the glial pruning pathway. To circumvent this therefore, we decided to examine the thermotaxis behavior defects of transgenic animals that over-express CED-10 only in AMsh glia. We reasoned that this cell-specific manipulation would avoid the complication of pleiotropic effects.

For cleaner behavior experiments, we integrated the extrachromosomal arrays (Ex) we had previously assayed (Figure 5B-E), and outcrossed the resulting integrated strains (Is). We then re-checked that the AFD NRE shape defects we had observed in the Ex lines persisted in these Is strains, and they did (SR, data not shown). We then examined these lines for thermotaxis behaviors at both 15^o^C and 25^o^C and found that both strains exhibited athermotactic responses. Strikingly, these responses are similar to those of *tax-2* mutants, which, like these strains, also have excess glia puncta. These results are now presented in Figure 7C-F and Figure 7-supplement 1C-D.

Thus, manipulating one component of the pruning pathway specifically in AMsh glia alters AFD-NRE shape and animal thermotaxis behavior. To our knowledge, the impact of glial engulfment on neuron shape or animal behavior has not been previously demonstrated with such single-cell resolution as we present here.

c. Several other places where conclusions need to be tempered or changed are listed below.

Each point is addressed below individually.

3. More details on statistical analysis are required. It's not always clear which tests were used for which sets of data, and there is no justification for using Kruskal-Wallis vs. Chi-squared tests. It's not clear whether corrections for multiple comparisons were carried out. In many of the figures, asterisks are placed in a way that makes it unclear which comparison they refer to.

Thank you for pointing this out. We have now changed our data presentation format, and hope this makes the comparisons referred to clearer. Further, prompted by this comment, we reconsidered carefully, and have chosen to use Fisher’s Exact Test on all proportional/binned data. While Chi Square is a less computationally taxing test, we have many conditions where bins have fewer than 5 samples in them. This means the assumptions of Chi Square are not met and therefore Fisher’s Exact Test is the best fit for our data. For statistical analyses of puncta quantifications, we believe One-Way ANOVA with multiple comparisons is the most appropriate.

4. The manuscript is rather difficult to read. The text repeatedly jumps back and forth between figures and panels, sometimes in a seemingly arbitrary way. Further, the text doesn't describe in sufficient detail exactly what is being shown in with regard to the many genetic manipulations carried out (alleles for example). The paper needs to be reorganized and rewritten to tighten the presentation of the data and make the story more accessible. It may also be useful to include a table summarizing all the candidate genes/pathways (including *C. elegans* and mammalian ortholog) and their phenotypes. Cartoon schematics to accompany the images would be useful for the non-*C. elegans* reader.

Thank you for pointing this out. To address this, we have now:

a. Significantly restructured the paper.

b. Paid heed that figure panels appear in the order that data is presented in the text.

c. Re-organized and/or split figures to present the data linearly.

d. Included the alleles tested in each figure legend.

e. Added schematics also in Figures 3 and 4 for easy reference.

[Editors' note: further revisions were suggested prior to acceptance, as described below.]

We find your resubmitted manuscript significantly improved, with nearly all of the reviewers' concerns addressed. The manuscript is reorganized, easier to follow, and the model is largely supported by the data. However, there are some remaining issues that need to be addressed.Many of these issues have to do with data interpretation – there are a number of cases in which alternative explanations are not ruled out, or where the data do not strongly support the conclusions drawn:1) Lines 133-135: The size ranges discussed here are not quite "in the range" of each other, as the text indicates. Please soften this conclusion.

Done. We have rephrased to “is of the same order of magnitude as”, in line with the previous statement that this is different from exophers.

2) Line 145-6: A trivial possibility that's not ruled out is that more puncta are seen in older animals because the AFD NRE isn't fully developed until the adult. Again, please soften your conclusion here or provide supporting data.

Agreed, and this is exactly what we also suggest. We have rephrased this to read “engulfment of AFD NREs by glia occurs after development of the AFD NRE is largely complete”. Hope this conveys the point better.

3) Lines 160 on: It's possible that no cilium fragments are seen in glia simply because the cilium is far smaller than the microvilli, which are larger and many. Please temper your conclusion.

We have made three edits to address this, and hope this rephrasing reads better.

a. We temper our conclusion to now say, “Data from both these approaches taken together suggest that the observed puncta in AMsh glia derive from AFD NRE microvilli as the primary, if not sole, source”.

b. That cilia fragments are too small to visualize was our concern too, until we found that *dyf-11*/ *osm-6* mutants have excess puncta. If cilia were a source, we should have seen less OR no change in puncta counts. Nonetheless, this comment made us realize that our quantification of these mutants is better placed in this section instead of in the activity section. We have now moved this.

c. We have also added a statement to interpret the quantification: “This indicates that cilia are likely not the primary source of glia puncta”.

4) Line 176: more than just the NRE is lost in ttx-1 mutants. One possibility could be that AFD no longer has the machinery to signal engulfment, possibly by putting PS on its outer leaflet.

We rephrased how we explain the *ttx-1* mutant, in addition to the edits noted in the point above.

We hope that together these addresses the concern. While we agree that *ttx-1* mutants are complex, here we interpret the phenotype in conjunction with four separate microvilli marked transgenic strains (tagged GCY-8, GCY-18, GCY-23, SRTX-1), time-lapse video microscopy of labeled microvilli fragments dissociating and moving into the glia (Video 1 and 2, Figure 3A, Figures 3D, 3E), and cilia studies. We suggest that when taken together, these strongly support the interpretation that microvilli are engulfed.

5) Lines 252-268: There are significant concerns about the interpretation of the results presented here, particularly the pat-2(RNAi) experiments. The text states that pat-2(RNAi) significantly blocks phagocytosis, but Figure 2G clearly shows that this is not the case. Further, many interaction tests are carried out using pat-2(RNAi) – any epistasis experiments using RNAi or non-null alleles need to be interpreted extremely cautiously, here and in the rest of the paper. This also leads to over-interpretation of the results of ced-1 experiments. Further, ced-1 hasn't been shown to function in AMsh, raising the possibility that its effects on puncta number could be indirect. As a result of these issues, the interpretations from this section don't seem particularly convincing or informative. This entire section could be removed without detracting from the advances made in your paper.

*pat-2* RNAi data: Our conclusion was based on 6 biological replicate RNAi experiments, that together show that *pat-2* RNAi significantly blocks phagocytosis. It is our error that while Figure 4 figure supplement 1 shows this data correctly, Figure 4G did not. Thank you for catching this. We apologize for this confusion and have corrected this.

Interpretation of RNAi data: We do not infer genetic epistasis with the RNAi experiments. Briefly:

i. *psr-1: pat-2* enhances a *psr-1* deletion mutant that removes all but 14 amino acids of the PSR-1 protein (Wang et al., Science 2003). Since *psr-1(tm469)* is likely a molecular null, and the double is stronger that *psr-1* alone, we think it is acceptable to infer that they act in parallel.

ii. *ced-1*: *pat-2* and *ced-1* have opposing phenotypes (decreased vs. increased engulfment, respectively) and *ced-1(e1754)* is a likely molecular null (Zhou et al., Cell, 2001). Further, we only infer from RNAi that the two genes do *not* act redundantly as PS-receptors. We think this conservative inference is acceptable but are open to alternate phrasing suggestion for CED-1 and PAT-2 roles as PS-receptors.

What we infer from our data is that CED-1/MEGF10 is *not* the primary PS-receptor driving glial engulfment of AFD-NRE. This, in itself, should be of significant interest for the glial pruning literature because as we note in the Discussion (Line 410-426), CED-1/Draper/MEGF10 is the major receptor for many instances of glial pruning in *Drosophila* and mammals. We therefore strongly favor reporting these findings in our manuscript. We think that other, indirect/non-PS-receptor, roles for *ced-1* will not take away from this conclusion and are also beyond the scope of this manuscript.

6) Lines 288-296: These experiments are difficult to interpret. The ced-10 phenotype is quite strong on its own, so floor effects get in the way of sorting out these genetic interactions. By themselves, these experiments don't convincingly show that ced-10 is downstream of ced-2/5/12 (though it may be legitimate to propose this based on previous findings). Similarly, the suppression of the psr-1 phenotype by ced-10 overexpression doesn't prove that ced-10 is downstream (it could act in parallel). Please soften your interpretation of these findings.

We have rephrased the *psr-1*/*ced-10* interpretation and hope this alleviates the concern. We now say, “Our data are consistent with the interpretation that, like in cell corpse engulfment, CED10/Rac1 GTPase likely functions in glia downstream of CED-2/CED-5/CED-12 and PSR-1, to promote AMsh glial engulfment of NREs.”

It is true that we base our interpretation of the *ced-2/5/12* – *ced-10* link partly on extensive prior literature in *C. elegans*, and also in *Drosophila* glial pruning (Zeigenfuss et al., Nat. Neuro 2012; Tasdemir-Yilmaz and Freeman, G and D, 2014). Indeed, this was our rationale for studying *ced-10*.

We agree that suppression of *psr-1* by CED-10 over-expression is not a sufficient experiment alone to establish that *ced-10* is downstream. We based this partly on established studies, and partly our own. We were careful to re-examine the *psr-1*/*ced-10* interaction in all genetic parameters originally used to place PSR-1 in the engulfment pathway (Wang et al., Science 2003). We tested (a) *psr-1* single and *psr-1; ced-10* double mutants, (b) cell-specific experiments placing both gene functions in the same cell, (c) rescue of *psr-1* with over-expressing CED-10. The additional confirmation Wang et al. had is Y2H with CED-5 (binds) and CED-10 (no binding).

Finally, while strong, *ced-10* mutant population phenotypes are non-zero (~50% animals have 1-9 puncta). Thus, we do not think this would have a floor-effect, and we should have seen noncanonical interaction, if at play.

7) Please consider reinterpreting the relationship between neural activity and engulfment. AFD temperature responses have not shown to be defective in osm-6 mutants (or provide reference), so the changes you see here may be unrelated to activity. Also, at 25C, AFD is believed to have higher activity, but your results show that there are more puncta at 25C than at 15C.

We see how this could have been confusing to read. We have rephrased this paragraph significantly (Line 321-328), hope this now reads better.

We agree that *osm-6* or *dyf-11* mutant data do not imply activity by themselves, and we do not say so. We only noted that these mutants were part of our logic to look closely at activity, when taken together with (a) ciliary proteins in thermotaxis (Tan et al., PNAS, 2007), and (b) localization of channels in cilia mutants (Nguyen, JCB 2012). Our activity inference is derived from *tax* mutants and the cell-specific chemo-genetic inactivation (His-Cl) experiments.